# The origin of the high electrochemical activity of pseudo-amorphous iridium oxides

Marine Elmaalouf[1,6], Mateusz Odziomek[2,6], Silvia Duran[3], Maxime Gayrard[2], Mounib Bahri [4], Cédric Tard [3], Andrea Zitolo [5], Benedikt Lassalle-Kaiser [5], Jean-Yves Piquemal[1], Ovidiu Ersen [4], Cédric Boissière[2], Clément Sanchez [2], Marion Giraud [1✉], Marco Faustini [2✉] & Jennifer Peron [1✉]

Combining high activity and stability, iridium oxide remains the gold standard material for the oxygen evolution reaction in acidic medium for green hydrogen production. The reasons for the higher electroactivity of amorphous iridium oxides compared to their crystalline counterpart is still the matter of an intense debate in the literature and, a comprehensive understanding is needed to optimize its use and allow for the development of water electrolysis. By producing iridium-based mixed oxides using aerosol, we are able to decouple the electronic processes from the structural transformation, i.e. Ir oxidation from $IrO_2$ crystallization, occurring upon calcination. Full characterization using in situ and ex situ X-ray absorption spectroscopy, X-ray photoelectron spectroscopy, X-ray diffraction and transmission electron microscopy allows to unambiguously attribute their high electrochemical activity to structural features and rules out the iridium oxidation state as a critical parameter. This study indicates that short-range ordering, corresponding to sub-2nm crystal size for our samples, drives the activity independently of the initial oxidation state and composition of the calcined iridium oxides.

[1] ITODYS, CNRS, UMR 7086, Université de Paris, Paris, France. [2] Sorbonne Université, CNRS, Collège de France, Laboratoire Chimie de la Matière Condensée de Paris, LCMCP, UMR 7574, Paris, France. [3] Laboratoire de Chimie Moléculaire (LCM), CNRS, Ecole Polytechnique, Institut Polytechnique de Paris, Palaiseau, France. [4] IPCMS, CNRS, Université de Strasbourg, Strasbourg, France. [5] Synchrotron SOLEIL, L'Orme des Merisiers, Saint-Aubin, Gif-sur-Yvette, France. [6] These authors contributed equally: Marine Elmaalouf, Mateusz Odziomek. ✉email: marion.giraud@u-paris.fr; marco.faustini@sorbonne-universite.fr; jennifer.peron@u-paris.fr

Hydrogen is expected to be a key player and to have a pivotal role in the future energy economy since, as an energy vector, it can be used as a fuel, but also as an efficient way to store energy at a very large scale.[1] The use of dihydrogen is however a sustainable solution for the environment only if $H_2$ is produced from abundant sustainable molecules such as water, using renewable energy sources such as the sun or wind. Among the water electrolysis technologies, Proton Exchange Membrane Water Electrolyzers[2] are expected to play a key role in particular for the coupling with intermittent conversion systems. One of the limitations of this technology is the high overpotential required at the anode for the Oxygen Evolution Reaction (OER) which directly impacts the price of hydrogen. To date, the only OER catalyst combining sufficient activity and stability is iridium oxide. Iridium-based materials including non-noble metals to form mixed oxides[3,4] or perovskites[5] are frequently proposed in the literature with higher electrochemical activity compared to pure $IrO_2$. These materials are usually prone to rapid degradation often resulting from the non-noble metal leaching and this, very interestingly, can sometimes lead to increased performance because of the modification of the iridium oxide structure. For a long time, it has been experimentally observed that iridium oxide materials presenting an amorphous or poorly crystalline (pseudo-amorphous) character (electrochemically synthesized or thermally obtained at low temperature) are more active than their crystalline counterparts (obtained after calcination at a higher temperature), but usually it is at the expense of their stability. From a fundamental point of view, tremendous efforts are currently being done for a better understanding of the structure-activity relationships[6–8] and the mechanisms of the electrochemical OER onto iridium oxide materials.[9–11] The development of advanced characterization techniques gives insights on both the actual electrocatalytic sites, the electrochemical reaction pathways, and the electrochemical species behavior during the OER. In particular, operando characterizations including X-ray absorption spectroscopy (XAS)[12–14] or near-ambient pressure X-ray photoelectron spectroscopy[10,15] are being intensively developed for mechanistic investigations even if they can still lead to contradictory interpretations. Despite extensive studies, the origin of the higher activity of amorphous materials is also still a matter of strong debate. The presence of hydroxo groups coordinated to iridium at the surface of highly active catalysts has been established.[16,17] On one hand, in recent papers, Strasser's group has proposed that the high activity of Ir-based material is related to the presence of Ir(III) or to a high OH concentration at the particles surface.[18] On the other hand, Willinger et al. have stressed the importance of a short-range order by preparing hollandite clusters stabilized by $K^+$ ions.[6] In the same trend, Sun et al. have proposed that a specific structural organization can lead to enhanced OER activity and have shown by DFT calculations that the distortion in $IrO_6$ geometry can optimize the binding strength between Ir-5$d$ and O-2$p$[19] and that increasing the lattice strain can increase the OER activity,[20,21] similarly to Pt catalysts for fuel cells.[22] Understanding the origin of iridium oxide activity and, in particular distinguishing the role of the oxidation state from the structural organization is of paramount importance since it also probably dictates the stability of the catalyst.[23]

In typical $IrO_2$ synthesis starting from $Ir^{3+}$ precursors, including solution synthesis or thermal treatment, the crystallization of the rutile structure occurs concomitantly with the oxidation of Ir(III) species into Ir(IV).[24] It is therefore very difficult if not impossible to decouple the oxidation of Ir species from the crystallization of the material, and it is thus not possible to discriminate unambiguously the impact of these factors on the electrochemical activity of the materials. We have recently proposed an aerosol synthesis route for the preparation of highly porous noble-metals and noble-metal alloys[25] as well as noble-metal pure and mixed oxides.[26] Tuning the crystallinity of the $IrO_x$ materials obtained through this method enables optimizing both activity and stability, and lead to materials outperforming many supported and non-supported Ir-based catalysts.[27] In this work, we are taking the advantage of the versatility of this green, cheap and scalable synthesis technique to prepare a large panel of porous iridium-non-noble metal mixed oxides and, in particular, iridium–molybdenum mixed oxides with tunable stoichiometry and crystallinity. By introducing molybdenum next to iridium into the structure, we are able to delay the crystallization of the materials while the oxidation of iridium (III) precursor into Ir (IV) occurs in the same temperature range as for pure iridium oxide. This unique behavior allows us to decouple the influence of the oxidation state from the structure of the material. Supported by a large set of techniques including surface XPS and bulk and operando XAS characterizations, we provide unambiguous evidence for the impact of the interatomic ordering onto the electrochemical activity of iridium-based materials towards the OER.

## Results

**Formation of iridium-based mixed oxides**. The general strategy to obtain porous iridium-based mixed oxides is illustrated in Fig. 1a for the case of Mo. The catalysts are prepared by templated spray-drying synthesis, starting from solutions containing $Ir^{3+}$ and $Mo^{5+}$ chlorides precursors with controlled Ir/Mo ratio, and PMMA beads as templating agent. With respect to typical solution synthesis based on nucleation and growth, the preparation of materials by spray-drying enables quenching of a metastable phase during fast drying in which the metallic precursors are intimately mixed. After spray-drying, $Ir_xMo_{(1-x)}Cl_y$/PMMA nanocomposite microspheres are obtained with a Ir/Mo ratio depending on the composition of the initial solution. A subsequent heat-treatment in air between 400 and 800 °C allows for the removal of the PMMA template (between 350 and 400 °C), decomposition of the chlorides and depending on the calcination temperature, crystallization of the inorganic phase.[26] The resulting highly porous Ir-based mixed oxides structure is exemplified in the TEM micrograph in Fig. 1a. This strategy was also extended to prepare a series of iridium-based mixed oxides including Mn, Co, Ti, and Mo (Fig. S1). Depending on the annealing temperature and on the composition of the material, the crystallization of the mixed oxide can be controlled as we will discuss it extensively later on.[26] Surprisingly, the crystallization and crystallite growth occur at the same temperature for pure iridium oxide and all the studied mixed oxides (around 450 °C) except for those containing Mo in which the crystallization is delayed (Fig. S1). The peculiarity observed in the case of $Ir_xMo_{(1-x)}O_x$ has driven us to pursue a deeper investigation of these materials. We have thus prepared a series of IrMo-based oxides with Mo contents of 10, 30, 50 and 70% annealed at different temperatures as well as pure $IrO_2$ obtained in the same conditions.

*Morphology of the catalysts*. SEM micrographs corresponding to $IrO_2$ and IrMo mixed oxides prepared with 10, 30 and 50% Mo illustrate this series of catalysts obtained after calcination at 500 °C. From 10 to 50% of Mo, SEM images show that the materials morphologies are very similar to pure $IrO_2$ and consist in porous microspheres with a broad size distribution, both features being characteristic of materials obtained by aerosol.[28] Each microsphere contains several cavities of ca. 250 nm which corresponds to the initial polymer beads diameter considering a slight contraction. However, while large nanocrystals can be observed on the surface of $IrO_2$ samples, no visible crystals are

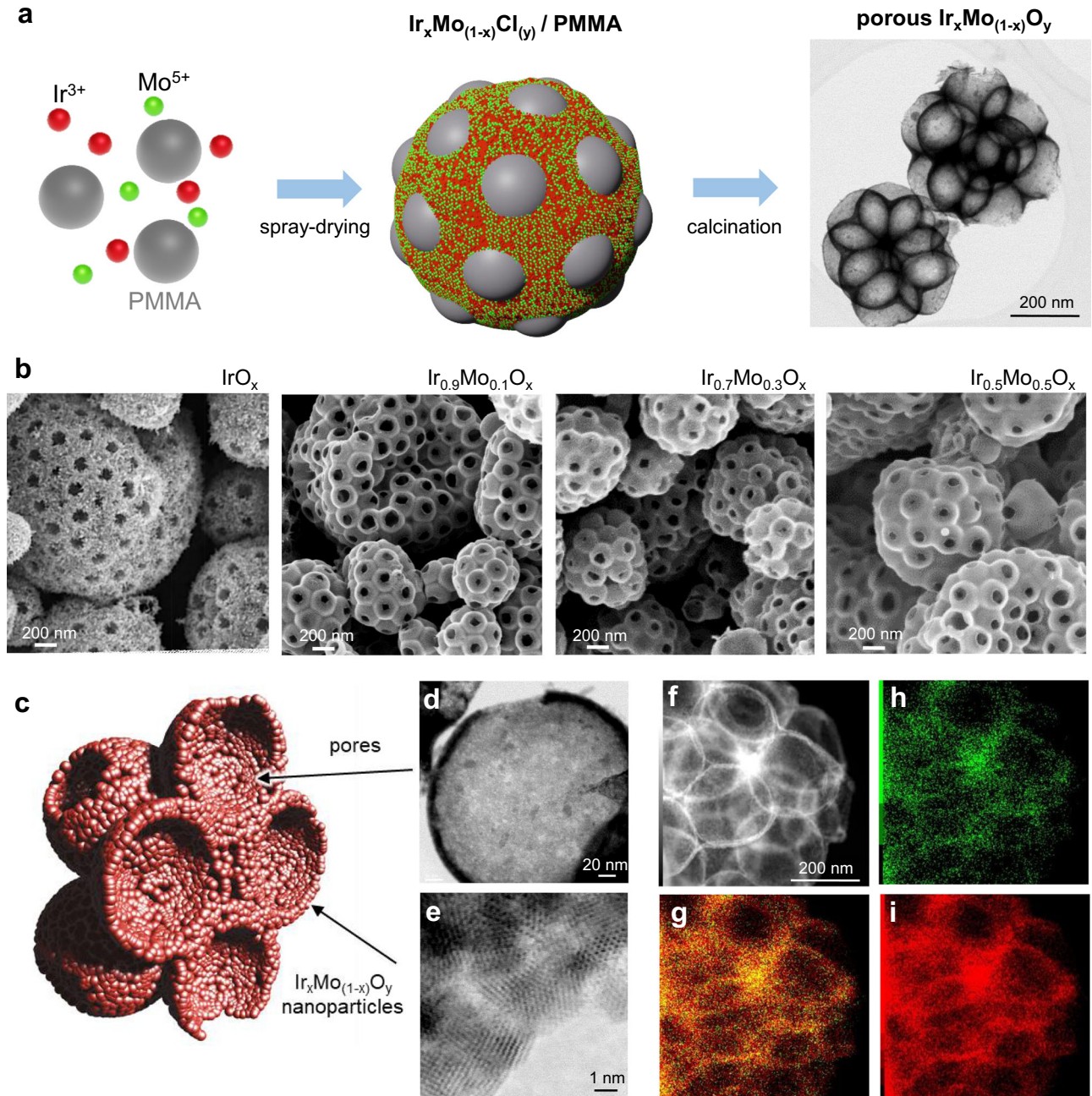

**Fig. 1 Synthesis and morphology of the porous catalyst. a** Schematic illustration of the synthetic process for the formation of porous $Ir_xMo_{(1-x)}O_y$ catalysts represented in the TEM micrograph (right side). **b** SEM micrographs sequence of $IrO_x$, $Ir_{0.9}Mo_{0.1}O_x$, $Ir_{0.7}Mo_{0.3}O_x$ and $Ir_{0.5}Mo_{0.5}O_x$ porous catalysts obtained at 500 °C. **c** Illustration of the morphology of a typical nanocrystalline catalyst. **d** and **e** HR-TEM micrographs of a single pore and of nanocrystalline wall respectively. **f** STEM image and corresponding EDX analysis of catalysts containing 30% of Mo and calcined at 550 °C. **h** Mo L edge (green), **i** Ir M edge (red), and **g** overlapping of the two elements.

seen on $Ir_xMo_{(1−x)}O_x$ compounds and a rather smooth surface is observed.

High-resolution transmission electron microscopy (HR-TEM) was then used to deeper investigate the nanostructure of mixed oxides catalysts. TEM micrograph in Fig. 1d represents a single pore of a catalyst prepared with 30 mol% Mo and calcined at 550 °C. The pore size is of ca. 250 nm. The walls of the porous microsphere are very thin, typically below 15 nm. For this sample, the HR-TEM micrograph in Fig. 1e highlights that the pores' walls are composed only of a few layers of sub-5 nm crystalline nanoparticles. The morphology of a typical mixed-oxide catalyst is thus illustrated in Fig. 1(c): the microspheres are highly porous

(typically 75% vol.[26]) composed of an interconnected macroporosity with thin walls that can be amorphous or composed by randomly oriented nanocrystals depending on the calcination temperature. This morphology is ideally suited for fundamental studies. The high accessibility of the porosity and the thin walls lead to a maximization of the Ir-based material involved in the catalytic reactions. As discussed in the next section, this will allow probing both surface and core Ir-based materials by bulk-sensitive techniques. These samples have allowed us to investigate the influence of the calcination temperature and composition on the morphology, structure, and electrochemical properties of these OER catalysts and elucidate some fundamental aspects of

the evolution of electroactivity of iridium-based oxides upon calcination.

To confirm the formation of mixed-oxides nanocrystals, materials structure and composition were probed by XRD, XRF, and TEM-EDX, respectively. Selected diffractograms of Ir-Mo samples prepared with 10, 30, 50% Mo and calcined at 550 °C, as well as that of $MoO_3$ are reported in Fig. S2. From 10 to 50% Mo, one single phase is observed with peaks characteristic of rutile $IrO_2$-type structure (ICSD 98-008-4577). The sample prepared with 70% Mo (Fig. S3) is made of two phases: $MoO_3$ and $Ir_{0.5}Mo_{0.5}O_x$ according to X-ray diffraction (XRD), and scanning electron microscopy coupled with energy-dispersive X-ray spectroscopy (SEM/EDX) analysis evidence large platelets containing only Mo and O along with the porous microspheres characteristic of Ir-Mo mixed oxides. For all the samples, the Mo content determined by Energy Dispersive X-Ray Fluorescence (EDXRF) corresponds very well to the initial stoichiometry of the precursors (Table S1). This is confirmed by TEM/EDX analysis of $Ir_{0.7}Mo_{0.3}O_2$ calcined at 550 °C which shows a homogeneous distribution of Ir and Mo throughout the microsphere pore walls (Fig. 1f–i). The formation of true iridium-molybdenum mixed oxides with a Mo content up to 50% is then achieved by the spray-drying method. In addition, as mentioned above, the presence of Mo delays the crystallization. This behaviour was probed by XRD for $IrO_2$ and all the IrMo mixed oxides prepared with 10, 30, and 50% Mo and calcined between 400 and 800 °C. As shown in Fig. S4, the crystallization temperature increases with the Mo content from 450 °C for pure $IrO_2$ to 550 °C for $Ir_{0.5}Mo_{0.5}O_x$.

*Electrochemical activity of IrMo-mixed oxides.* The activity toward the OER of IrMo-mixed compounds prepared with 10 to 50% Mo was then evaluated using a three-electrode set-up (see "Methods"). For each Mo content, the activity of the materials calcined at 400, 450, 500, 550, 600, and 800 °C was measured (Fig. S5) and the current density values recorded at 1.5 V are plotted vs. calcination temperature for each sample on Fig. 2a.

It is well known that for pure iridium oxide, the calcination temperature strongly influences its electrochemical performance, lower calcination temperature leading to more active materials but often at the expense of the stability. Confirming these observations, in the case of pure $IrO_2$ porous microspheres, we have shown that the most active catalysts were those calcined at 400 and 450 °C, higher calcination temperature lowering the performance.[26]

Surprisingly, in the case of IrMo mixed materials, the highest performance does not systematically correspond to the lowest calcination temperature but it is shifted toward higher temperatures as the Mo content in the material increases. For most of samples, the Tafel slopes range from 70 and 90 mV/dec with a typical value of 70–74 for the most active materials of each series (SI6). The activities of the most active materials, i.e. obtained at the optimal calcination temperature, for each Mo content can be compared on the bar diagram reported in Fig. 2b. At the optimal calcination temperature, the activity of the various materials decreases with increasing Mo content, but the activity reported to the actual mass of iridium remains almost constant. This suggests that Mo does not affect the intrinsic activity of iridium but its presence modifies the temperature at which iridium oxide is the most active. Instead, the most active catalysts of each series are obtained around the temperature of crystallization, more amorphous or more crystalline catalysts of each series being systematically less active. The origin of the electroactivity of these pseudoamorphous (or less crystalline) materials can be attributed to several parameters that will be investigated in detail and discussed in the following sections.

*Possible influence of Ir oxidation state.* Previous literature suggested that the Ir oxidation state of the catalyst might play a crucial role on its activity.[7] In order to verify the possible role of Ir oxidation state on the evolution of the activity of IrMo mixed oxides, we have probed the iridium surface oxidation state by XPS and followed the evolution of iridium bulk oxidation state analyzing the X-ray absorption near edge structure (XANES) spectra both ex situ on pellets made with catalysts powder and in situ in a electrochemical cell under electrolysis conditions. In addition, we have followed the structural evolution of the materials and probed the local environment of iridium analyzing the extended X-ray absorption fine structure (EXAFS) spectra.

We first analyzed the Ir oxidation state of the initial catalysts in Fig. 3a, the most electroactive materials for each composition are highlighted by the dotted-red line. Both from XPS measurements and XANES spectra analysis, we observe that the transition of iridium (III) into iridium (IV) occurs at the same temperature whatever the Mo content. In addition, iridium oxidation at the surface (XPS) and in the bulk (XANES) occurs quasi-simultaneously with a slight delay for a complete oxidation of the bulk (450 °C vs. 500 °C). The Ir XPS spectra recorded after calcination at various temperatures for each Mo content are

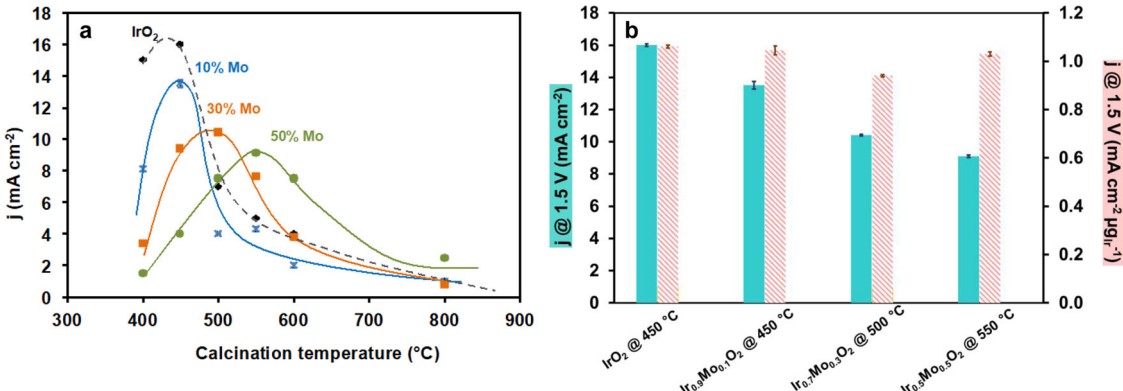

**Fig. 2 Electrochemical activity of the materials. a** Current densities values recorded at 1.5 V vs. NHE for pure $IrO_2$ (diamonds, black) samples prepared with 10% (crosses, blue), 30% (squares, orange), and 50% (circles, green) of Mo as a function of temperature of calcination—lines are guide for the eyes. SD error bars calculated from measurements performed on four different deposited inks are included and represented in black for each point; **b** Current density recorded at 1.5 V vs. NHE for each sample at the optimal calcination temperature. The current density is reported to the geometrical surface of the electrode (plain bars) or to the mass of iridium calculated using the stoichiometry of the mixed oxide (dashed bars). SD error bars included. Source data are provided as a data source file.

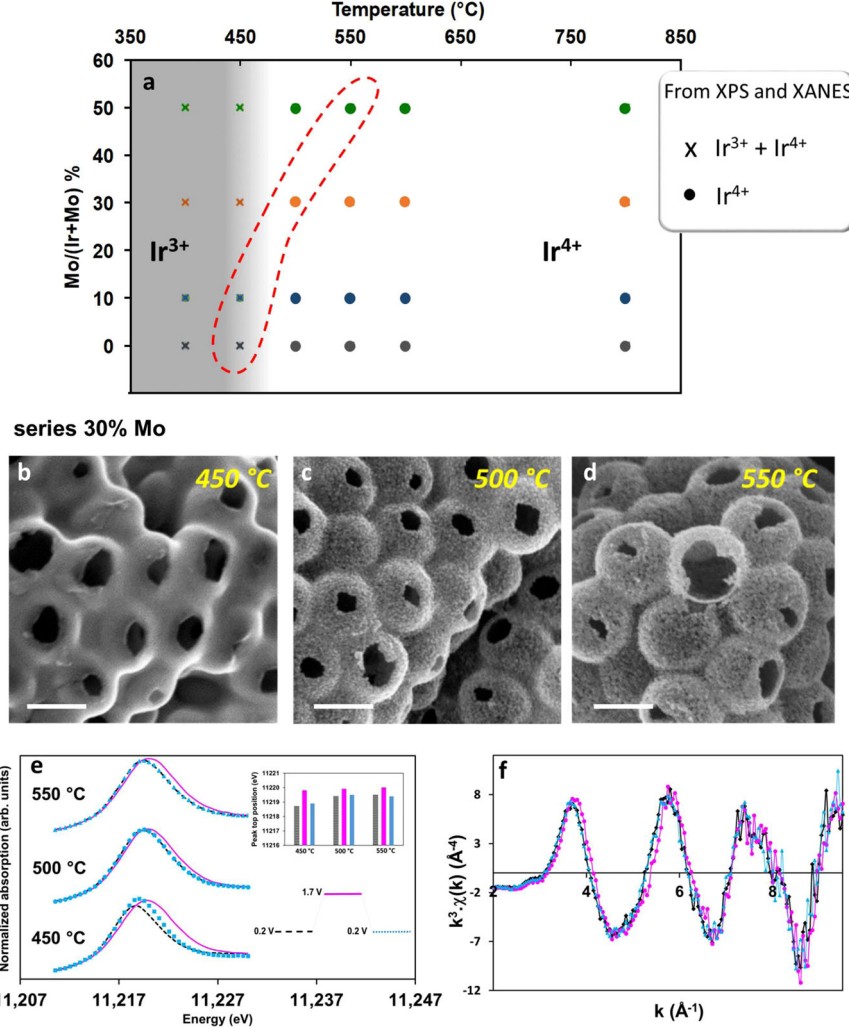

**Fig. 3 Initial iridium oxidation state does not drive electrochemical activity. a** Graph summarizing the main oxidation state observed from XPS spectra and XANES region spectra analysis for the different Mo content (theoretical %) as a function of calcination temperature. **b**–**d** SEM images of the series prepared with 30% Mo and calcined at 450 °C, 500 °C and 550 °C, scale bar 200 nm, and corresponding operando characterization at the Ir L₃-edge: **e** XANES spectra recorded at 0.2 V vs. NHE, 1.7 V vs. NHE and back to 0.2 V vs. NHE and the corresponding peak-top values; **f** Superposition of the EXAFS spectra of the 30% Mo series (calcined at 500 °C) at 0.2 V vs. NHE (black diamonds), 1.7 V vs. NHE (pink circles) and back to 0.2 V vs. NHE (blue triangles). Source data are provided as a data source file.

displayed in Fig. S7. The peaks characteristic of Ir $4f_{7/2}$ and $4f_{5/2}$ centered at 62.3 and 65.3 eV (typical of Ir(III)) at 400 °C shift to 61.7 and 64.7 eV (typical of Ir(IV)) at 450 °C and remain at the same position as the temperature increases. The transition from Ir (III) to Ir(IV) happening between 400 and 450 °C is accompanied by the decrease in the intensity of O 1s peak located at 531 eV (characteristic of O in metal-(OH) bonds) and the increase in the O 1s contribution located at 530 eV characteristic of metal–oxygen bond in oxides (Figure S8). It is noteworthy that the ratio of the M(OH)/MO peaks does not change significantly when increasing the calcination temperature. The transition from Ir(III) into Ir(IV) is also observed looking at the position of either the peak top position or the absorption edge value of the XANES spectra, but at a slightly higher temperature; a shift characteristic of Ir(IV) is observed at 500 °C and above (Figs. S9 and S10). The diffusion of the oxygen in the bulk of the particles for the complete oxidation of the materials is probably responsible for these slight differences. For all the samples, including pure IrO₂, we observe that iridium only exists as Ir(IV) above 500 °C, whereas the most active electrocatalysts containing 30 and 50% of molybdenum are those calcined at 500 °C and 550 °C,

respectively. While we cannot exclude that the initial oxidation state of iridium, and especially the presence of Ir(III), (or a higher initial OH amount at the material surface) has an influence on the activity, these results suggest that this is not the predominant parameter to explain the origin of the most active electrocatalysts. The initial Ir oxidation state (measured ex situ) of the as-prepared catalysts is a very important parameter when designing a catalyst and it is routinely reported by the community working on synthesis of Ir-based materials for OER. However, it has been shown that the Ir oxidation state can increase at high potential,[10,12,29] and one could argue that Ir oxidation state can evolve irreversibly during the OER, raising doubts about our conclusions on the minor role of the Ir oxidation state of the initial catalyst.

All the above mentioned discussion is based on the initial chemical and structural state of the Ir-based catalysts investigated by ex situ analysis. To gain a better understanding, we have performed in situ XAS experiments at Ir L₃-edge in an electrochemical cell and followed the evolution of both the oxidation state and the material structure during operando measurements for the series of catalysts prepared with 30% Mo

and calcined at different temperatures (450, 500, 550 °C). This series was chosen because all the materials have similar Mo content but different Ir initial oxidation state. The SEM micrographs of the three samples are shown in Fig. 3b–d. All the materials have similar morphology, but while the sample calcined at 450 °C is amorphous, the samples calcined at 500 and 550 °C exibit crystalline pore's walls. As discussed above, the peculiar porous morphology with thin pore's wall enables maximizing the Ir atoms participating in the reaction. We have thus followed Ir oxidation state during OER by in situ XAS experiments.

XANES spectra of the materials recorded at various potential and corresponding peak top values are plotted on Fig. 3e. The initial XANES peak top positions measured in the electrochemical cell at 0.2 V vs. NHE confirmed ex situ measurements: the oxidation state is lower (corresponding to Ir(III) + Ir(IV)) for materials calcined at 450 °C and increases to Ir(IV) for materials calcined at 500 and 550 °C. At 1.7 V vs. NHE, under OER conditions, as expected, the oxidation state of all the materials increases. More importantly, whatever the initial oxidation state, all of them tend to have a similar oxidation state during OER (>Ir (IV)). This higher oxidation state (formally > +4) observed at high potential is in agreement with results from the literature where it has been shown that the Ir oxidation state can increase at high potential.[10,12,29] Since the XAS experiment probes all the Ir atoms of the material, this significant increase in XANES peak top position is comparable to previous in situ studies[12] and is an additional evidence that the reaction is not occurring only at to outer surface of the micrometric spheres. Since the microspheres are highly porous, the catalytic reaction takes place also in the inner surface of the sphere. Coming back to 0.2 V vs. RHE, all the materials recover their initial oxidation state, meaning that OER conditions did not affect the electronic structure of the material permanently. To assess the material structure evolution during OER, we have analyzed the Ir EXAFS spectra of the most active material of the series (calcined at 500 °C) and recorded prior-to, during and after OER conditions (Fig. 3f). Although the data quality obtained operando allows to analyze $k$-values only up to 9 Å$^{-1}$, we can see that the spectra recorded prior-to and after OER perfectly overlap; this observation rules out a strong evolution of the local order around Ir or of the material structure in OER conditions over the course of the experiment. Indeed, considering the results of the in situ experiments, we can emphasize that whatever the initial oxidation state of our catalysts, during OER the Ir oxidation state increases toward similar values. This implies that the differences in electrochemical activity measured for the 30% Mo series (calcined at 450, 500 and 550 °C) do not depend on the oxidation state during OER. In addition, the reversibility of the process indicates that there is no major structural or chemical changes after OER. The experiments performed both ex situ and in situ indicate that the electroactivity trend shown in Fig. 2a cannot be correlated with initial iridium oxidation state, nor the one measured operando.

**Possible influence of Mo and Cl**. Since we are comparing pure iridium-based materials with iridium-molybdenum mixed compounds, we need to investigate as well a particular effect of Mo onto Ir electronic structure that could strongly impact Ir electronic structure and therefore the material performance. Similarly, since chloride precursors were used for both Ir and Mo, we have probed the chlorine content for each series of materials as a function of temperature and looked if a correlation between Cl and electrochemical activity of the materials could be found. As observed on Mo XPS spectra (Fig. S8), Mo chemical transformation occurs at the same temperature whatever the Mo content

in the material. For the three series of materials containing 10%, 30%, or 50% Mo, at 400 °C, Mo(VI) is observed at the materials surface and it is reduced into Mo(V) between 400 and 500 °C. It remains in the same oxidation state up to 600 °C, while at 800 °C, a small contribution of Mo(IV) is observed along with Mo(V) (Fig. S8). The overlapping of Ir XANES edges and peak top positions (Fig. S10), as well as Ir 4f peaks positions seen on XPS spectra of the samples with different compositions but calcined at a similar temperature, also rules out a strong electronic effect of Mo onto Ir. For all the samples prepared without Mo and with 10, 30, and 50% Mo we have calculated the average number of Cl and O in the coordination sphere of Ir from EXAFS spectra and determined the chlorine content as a function of the calcination temperature from XPS experiments; values are reported in Tables S2 and S3, respectively. Temperature-resolved Fourier transform (FT) of the Ir L$_3$-edge EXAFS spectra for pure Ir-sample and for the sample prepared with 30 and 50% Mo upon calcination are shown in Figure S11. Between 400 and 450 °C, the Ir-Cl peak at R' = 1.99 Å (uncorrected for phase shift) disappears concomitantly with the appearance of the characteristic Ir-O peak at R' = 1.65 Å.[25] From EXAFS fitting, at 400 °C, the average number of Cl neighbors is higher than the oxygen one (ca. 3.7 vs 2.1). At 450 °C, the trend reverses, and the number of O neighbor is predominant. It then reaches a constant value up to 800 °C. Whatever the Ir/Mo ratio, chloride content varies similarly. No distances characteristic of Ir–Cl bond are observed on EXAFS spectra above 450 °C. From XPS measurements, we also observe that Cl content strongly decreases from 400 °C to 450 °C (Table S3), and then reaches a constant value from 500 °C up to 800 °C of ca. 0.05 for the ratio Cl/(Ir+Mo). The main conclusion is that highly active materials are obtained around the crystallization temperature with and without Cl. The most active materials prepared without Mo and with 10% of Mo contain a rather high amount of residual chlorine. Inversely, the material prepared with 30 and 50% of Mo are converted into corresponding oxides (without Cl) at 500 °C, while most active materials are observed at 500 °C and 550 °C for these two series, respectively. Therefore, the presence of chlorinated species cannot be accounted for explaining the higher/lower activity of the catalysts.

**Influence of surface area and crystallinity**. Since chemical transformations occurring upon calcination cannot be accounted for the evolution of the electrochemical activity, we have looked at the material surface area and structure. XRD results are summarized in Fig. 4a where the crystallite size, determined from Rietveld refinement when possible, is depicted for each composition and each calcination temperature (XRD diffractogramms of all the samples are given in Fig. S4). We first observe that the introduction of Mo into the mixed oxide delays the crystallization temperature and while the sample prepared with 10% Mo is already crystallized at 450 °C, typical peaks from the rutile structure only appear at 550 °C for the sample prepared with 50% Mo. The other important observation is that for the same calcination temperature, smaller crystallites (broader peaks) are obtained with higher Mo content (Fig. S2). This is observed in the whole range of temperature studied, and this decrease is more pronounced as the Mo content in the mixed oxide increases. At 600 °C, the crystallite size is of 15.8 nm for pure IrO$_2$, 6.5 nm with 10% of Mo, 4.5 nm with 30% of Mo, and of only 2.5 nm for the sample prepared with 50% of Mo. The position of the (110) peak shifts towards low 2θ values as the Mo content increases in the IrMo-mixed oxide while the (011) peak shifts toward higher 2θ. This is indicative of an increase in the $a$ unit-cell parameter and a concomitant decrease in the $c$ parameter as illustrated in Fig. S12.

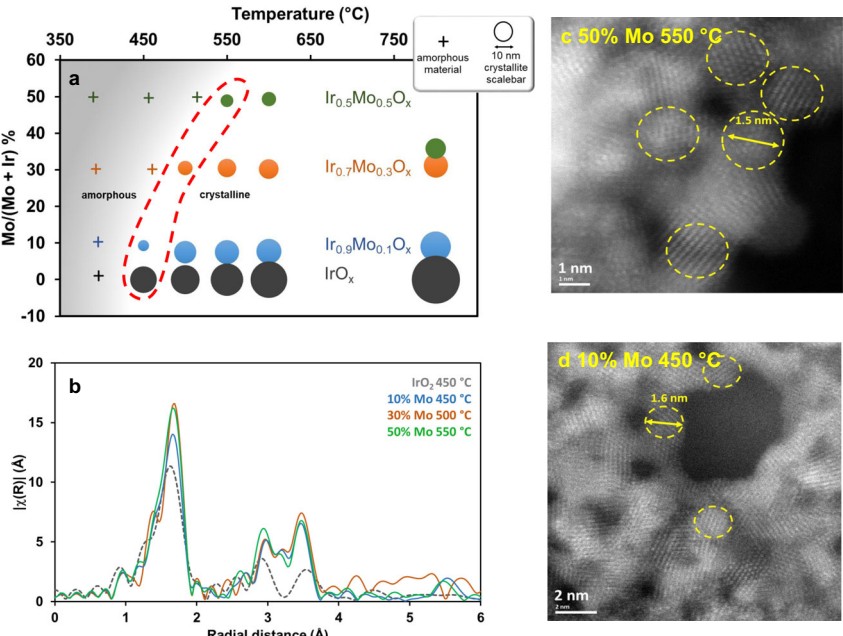

**Fig. 4 Most active materials present similar structural features. a** Graph summarizing X-Ray Diffractograms analysis and chemical composition determined by EDXRF: amorphous samples are labeled with crosses (+), crystalline samples are marked with plain circles (●) and the circle diameter is proportional to the crystallite size determined by Rietveld analysis, the position of each symbol in the y axis corresponds to the Mo content determined from EDXRF analysis. The most electroactive materials for each composition are highlighted by the dotted-red line. **b** Superposition of the FT-EXAFS spectra recorded for the most electrochemically active materials of each Mo content i.e. pure $IrO_2$ calcined at 450 °C, $Ir_{0.9}Mo_{0.1}O_x$ calcined at 450 °C, $Ir_{0.7}Mo_{0.3}O_x$ calcined at 500 °C and $Ir_{0.5}Mo_{0.5}O_x$ at 550 °C. STEM-HAADF images of the samples prepared with **c** 50% Mo calcined at 550 °C and **d** 10% Mo calcined at 450 °C. Source data are provided as a data source file.

From the peak width, we can see that the particles initially grow along the $c$ direction and since the $c$ parameter is smaller than $a$, the particles present a rather isotropic shape as confirmed by TEM micrographs (Fig. 4c, d). As a consequence, the morphology of the $Ir_xMo_{1-x}O_y$ particles differs drastically from that of pure $IrO_2$ (Fig. S11). While pure $IrO_2$ crystallizes into strongly anisotropic structures (typically nanoneedles),[26] isotropic nanoparticles compose the walls of the porous microspheres obtained with Ir-Mo mixed oxides. Within each series containing Mo, $a$ and $c$ parameters only weakly vary with temperature (Table S4). This also suggests that no variation in lattice strain is expected with the temperature for the materials with the same composition.[20]

Looking at the area highlighted by the red-dotted line in Fig. 4a, we observe that the most active materials present similar structural features. For each composition, the most active material is not amorphous but it is the one presenting the smallest crystallite size (along the separation line between amorphous and crystalline materials). In this area, with such small particles and very broad peaks, a better insight of local order around iridium atoms can be obtained by analyzing EXAFS spectra at different stages of the formation of the materials. Temperature-resolved Fourier transform (FT) of the Ir $L_3$-edge EXAFS spectra for pure Ir-sample, samples prepared with 30 and 50% Mo upon calcination are shown in Fig. S11. For all the materials, as mentioned above, between 400 and 450 °C, the Ir-Cl peak disappears concomitantly with the appearance of the characteristic Ir-O peak.[25] As the temperature increases, the peaks characteristic of Ir-Ir interaction appear at $R' = 2.91$ Å and 3.55 Å and grow as the calcination temperature increases due to the crystallization of the material.[16,30,31] However, this peak, characteristic of Ir-Ir interaction, appears at different temperatures depending on the Mo content. As the Mo content increases, the temperature at which the peak appears increases. For the

sample prepared with 50% of Mo, below 550 °C (optimal calcination temperature), only the peak corresponding to the first shell of oxygen atoms is visible, which is characteristic of an amorphous structure. Above this optimal temperature, two peaks characteristic of Ir-M (M = Ir, Mo) interactions (observed at 2.97 and 3.60 Å) increase in intensity which indicates a longer-range ordering. The presence of the additional peak at 3.11 Å, not observed for pure $IrO_2$, strongly supports the other analyses on the presence of Mo in the rutile lattice of Ir-based oxide. Figure 4b displays the FT-EXAFS spectra of the most active catalysts for each Mo content (i.e., the optimal calcination temperature for each Ir/Mo ratio). Importantly, all the spectra, including the one for the most active pure iridium oxide calcined at 450 °C, present strikingly similar features i.e. no peaks typical for long-range ordering and the intensity of second and higher shells peaks much lower than that observed for bulk with respect to the intensity of the first-neighbors peak. As observed and calculated for metals,[32] this strongly indicates that the clusters of $IrO_x$ are so small that the average number of neighbors of M absorbers in the second shell differs drastically from the expected values in bulk crystalline $IrO_2$. HR-TEM analysis was performed on the most active materials prepared with 10 and 50% of Mo and calcined at 450 °C and 550 °C, respectively (Fig. 4c, d). In both cases, small crystalline domains below 2 nm can be observed which confirms the structural similarities between the most active materials, regardless of the composition.

To get a better insight on the evolution of the materials, physisorption measurements were performed for each sample. Since the ECSA cannot be detemined reliability for Ir-based oxides (not shaped as model films), BET surface area was determined. The evolution of specific surface by BET is a good indicator to verify whether the increase or decrease of activity can be due to an evolution of surface area of the material. It is also worth reminding that the Ir/Mo ratio in the surface and in the

bulk is similar, ruling out a possible Ir segregation on the surface. Data are presented in Fig. S14. In each series, we observe that the surface area increases until the crystallization temperature is reached. Indeed the beginning of the crystallization corresponds to formation of small nanoparticles characterized by a high surface area. Increasing the calcination temperature leads to crystal growth that is usually accompanied by a decrease of surface area. This decrease is clearly observed for the series prepared without Mo and with 10% Mo characterized by larger crystal sizes. However, the surface area of materials prepared with 30 and 50% does not vary and stabilize at ca. 70 m$^2$ g$^{-1}$; this is probably due to the formation of mesopores within the walls (BET isotherms in Fig. S15) which counterbalances the decrease in surface area that should be observed due to particle growth. While it is not straighforward to establish a direct correlation, the increase of the activity between amorphous and crystalline materials could be attributed to an increase in surface area. On the other hand, the decrease in activity observed between very small crystallites and larger ones unlikely depends on surface area that remains almost constant for higher Mo content showing small particle sizes.

## Discussion

As shown above, the most active materials of each series (calcined at higher temperature for higher Mo content) present different iridium and molybdenum oxidation states but similar structural features. These results show that the electrochemical activity of the material is independent on the initial Ir oxidation state. The in situ experiment is also strengthening our conclusion on the predominant role of the structure to explain the trend of the electrochemical activity of the samples since all the materials present similar oxidation state in OER conditions.

In addition, the most active materials are not fully amorphous, their X-ray diffractograms present broad peaks indicative of the presence of very small crystallites, and optimal intensity of peaks characteristics of Ir-Ir distances are observed in the FT-EXAFS spectra. It is very important to point out that for pure iridium oxide, this degree of structural order (ultra-small crystallite size) is usually reached upon calcination at the temperature corresponding to the transition of Ir(III) into Ir(IV) making it is almost impossible to disentangle the effects of the structure from that of the oxidation state ones. In line with work of Guan et al.[33] we show that ultrasmall IrO$_2$ nanoparticles (below 2 nm) show the highest activity. The peculiar crystallization mode of IrMo mixed oxides compared to pure IrO$_2$ or other mixed oxides allows decoupling oxidation and crystallization processes occurring upon calcination, therefore discriminating the influence of these different parameters onto the electrochemical activity of the materials. In light of these results, this paper is much in favor of the influence of a particular structural organization and a short-range order to account for the high activity of iridium oxides calcined at low temperature than of the impact of iridium oxidation state.

Regarding the structure, we observed that the highest electrochemical activity is obtained for the smallest crystallites in each series. This cannot only be related to an increase in surface area. Combining isotope labeling and atom probe tomography, Schweinar et al. have shown recently that during OER, even in the case of rutile IrO$_2$, O atoms could be exchanged between the oxide lattice and water down to 2.5 nm below the surface, and this depth could be even higher in the case of amorphous materials.[34] If this is the case, this would mean that for ultrasmall particles, all the atoms could participate in the catalytic reaction, making surface area a less relevant parameter. However, recent results from the literature suggest the importance of the short-range

order as reported by Willinger et al. for K-IrO$_x$ materials.[6] This assumption is also supported by comparing our findings with EXAFS spectra reported in the literature,[16,35] and this despite various synthesis procedures or even different types of materials. For instance, Gao et al. recently reported highly active Li-IrO$_x$ "amorphous" materials, for which the EXAFS spectra show features very similar to our most active materials, i.e. indeed no long range order but a weak signal for short range Ir-Ir distance.[35] Abbott et al.[16] also reported very active iridium oxide synthesized at 350 °C from Ir(acac)$_3$ using a modified Adams fusion method, consisting of 1.7 ± 0.4 nm particles with a specific surface area of 150 m$^2$ g$^{-1}$ and the corresponding EXAFS spectrum is also very similar to those presented in Fig. 4b. This short-range order could also account for the enhanced activity observed in "leached-mixed oxides". For instance, Reier et al. proposed that Ni leaching from IrNi-mixed oxides acts as promoters for the formation of reactive OH groups, reducing at the same time the quantity of inactive oxide surface termination.[17] They hence attributed the high activity to the high number of OH groups. Actually, the proposed scheme for the resulting surface considers small metal oxide clusters with an increased number of edges and corner atoms, very similar to what we currently propose.

In conclusion, using aerosol synthesis, we have been able to obtain a wide range of binary iridium-based metal oxides with tunable stoichiometry and crystallinity. Among them, IrMo-mixed oxides present a unique behavior, which allows decoupling oxidation from crystallization processes occurring upon calcination. A wide range of characterization techniques, including operando XAS, were used for a systematic analysis of the materials structure and chemical composition as a function of composition and calcination temperature. Using these data, we unambiguously evidenced the role of the material structure on its OER electrochemical activity, whatever its initial oxidation state. We, therefore, propose that the high electrochemical activity of iridium-based oxides calcined at low temperature is unlikely due to the iridium oxidation state, but rather strongly depends on the material structure. In addition, within crystalline materials, we show that smallest particles lead to increased activity. Besides these important findings, this study showcases that aerosol synthesis combined with controlled thermal treatment represents an ideal playground for material synthesis with tunable composition and offers promising perspectives in terms of applications as well as fundamental studies in the field of electrocatalysis.

## Methods

**Chemicals**. IrCl$_3$,$x$H$_2$O (98%) was purchased from Alfa-Aesar and MoCl$_5$,$x$H$_2$O (98%) from Sigma-Aldrich. PMMA beads were synthesized by radical polymerization from methyl methacrylate following the protocol described in ref. [11,36] wt% dispersion of 300 nm PMMA beads in water was used for the hybrid materials preparation. Commercial iridium (IV) oxide, Premion®, 99.99% (metals basis) and Ir were purchased from Alfa-Aesar.

**Catalyst preparation**. In a typical synthesis, IrCl$_3$,$x$H$_2$O/MoCl$_5$,$x$H$_2$O (2.77 mmol of Ir+Mo) mixture was dissolved in water (27 ml) under stirring over 30 min. The 5.5 g of 11 wt% polymer dispersion was then added to the metal-containing solution to obtain a final polymer/metal precursor ratio of 0.22 g/mmol. After 10 min stirring, the solution was spray-dried with a B-290 atomizer from Büchi. The spray-dryer parameters were set as follows: input temperature: 220 °C, pump flow: 5 ml min$^{-1}$. The recovered powder was then calcined at a temperature comprised between 400 and 800 °C for 15 min under static air. The temperature ramp from ambient to the desired temperature was of 2 °C min$^{-1}$.

**Scanning electron microscopy (SEM)**. SEM-FEG images were obtained using a Zeiss SUPRA 40 FESEM, operating at 5 kV.

**Transmission electron microscopy (TEM)**. TEM analyses in the TEM and scanning TEM (STEM) modes were carried out using a JEOL 2100 FEG S/TEM microscope operated at 200 kV equipped with a spherical aberration probe corrector. Before analysis, the samples were dispersed in ethanol and deposited on a

holey carbon-coated TEM grid. In STEM, the images were recorded using a high-angular annular dark field (HAADF) detector with inner and outer diameters of about 73 and 194 mrad. Energy Dispersive X-ray Spectroscopy mapping was performed using a JEOL Silicon Drift Detector (DrySD60GV, sensor size 60 mm$^2$) with a solid angle of ~0.6 srad.

**X-ray diffraction (XRD)**. XRD measurements were carried out using a Panalytical X'pert pro diffractometer equipped with a Co anode ($\lambda K_\alpha = 1.79031$ Å) a multichannel X'celerator detector. All the diffractograms were fitted via MAUD program (Material Analysis Using Diffraction), a general X-ray diffraction program based mainly on the Rietveld refinement method,[37] and allowing determination of $a$ and $c$ parameters as well as the calculated crystallites sizes for each diffraction peak.

**Energy dispersive X-ray fluorescence (EDXRF)**. Elemental analyses of the powders were conducted by EDXRF using an epsilon 3XL spectrometer from Panalytical equipped with a silver X-ray tube. The calibration was performed by depositing a mass in the range 5–20 μg of the standard solution of each element on a polycarbonate membrane. The same conditions were adopted for all samples. The detection limits for Ru and Ir were determined to be 3.7 and 12 ng, respectively.

**X-ray photoelectron spectroscopy (XPS)**. XPS spectra were recorded using a K-Alpha+ spectrometer from Thermofisher Scientific, fitted with a microfocused, monochromatic Al K$_\alpha$ X-ray source ($h\nu = 1486.6$ eV; spot size = 400 μm). The pass energy was set at 150 and 40 eV for the survey and the narrow regions, respectively. Spectral calibration was determined by setting the main C1s (C–C, C–H) component at 285 eV.

**Physisorption**. Physisorption studies were performed with N$_2$ at 77 K using a Belsorp-max apparatus from MicrotracBEL. Before analysis, the samples were outgassed at 423 K for 12 h under 0.1 Pa. The BET processing was carried out in the relative pressure range 0.05–0.25.

**Electrochemical characterization (OER protocol)**. The potentiostat used for cyclic voltammetry was an Autolab PGSTAT 12, and ohmic drop was compensated in all cases (typically 30 Ω). The working electrode was a 5 mm diameter glassy carbon rotating disk electrode (Pine Instrument), carefully polished and ultra-sonically rinsed in absolute ethanol before use. The counter electrode was a platinum wire and the reference electrode an aqueous saturated calomel electrode. All experiments were carried out in air at 10 mV/s and at a rotating rate of 1600 rpm. A 0.05 mol L$^{-1}$ sulfuric acid solution (sulfuric acid, 0.1 N standardized solution, Alfa Aesar) was used as supporting electrolyte. From a mixture of 1 mg of synthesized material or iridium oxide (99.99%, Alfa Aesar), 2 mg of carbon Vulcan XC72R (Cabot), 250 μL of Nafion solution (5% w/w, Alfa Aesar), and 250 μL of deionized water (0.059 μS cm$^{-2}$), 8 μL of this suspension were deposited on the electrode surface, which was then dried in air and left for 30 min at 60 °C in an oven. Catalysts were then submitted to 50 cycles between −0.2 and 1.2 V at 200 mV s$^{-1}$, and the 50th forward scan recorded at 10 mV/s in 0.05 M H$_2$SO$_4$ is presented.

**X-ray absorption spectroscopy (XAS)**. Ir L$_3$-edge X-ray absorption spectra were collected in transmission mode at the SAMBA beamline (Synchrotron SOLEIL) at a ring energy of 2.75 GeV and a nominal current of 500 mA. The energy was monochromatized using a sagittally focusing Si 220 monochromator and two Pd-coated mirrors that were used to remove X-rays harmonics. Each sample was dispersed in boron nitride with a 16 wt% proportion (to get a suitable Ir L$_3$-edge jump) and the resulting powders were processed as 3 mm diameter pellets. Data reduction and analysis was performed using the Athena software.[38] The XAS data were binned, summed, and normalized using Athena, a program in the IFEFFIT package.[38] The absorption edge E$_0$ was measured as the first inflection point in the derivative spectrum and calibrated vs. an Ir foil. The XANES and EXAFS spectra were normalized from 42 to 115 eV below the edge and 150 to 1470 eV above the edge, respectively.

**Operando XAS**. Ir L$_3$-edge X-ray absorption spectra were collected in fluorescence mode at the SAMBA beamline (Synchrotron SOLEIL, see characteristics below). Cyclic voltamograms and constant potential electrolysis procedures were performed using a Bio-logic SP-300 potentiostat, connected to a custom-made electrochemical cell suitable for in situ/operando XAS measurements. The cell is made of a single, square-shaped glass compartment (3 × 3 cm), fitted at the top with a cylindrical screw cap (4.5 cm inner diameter). A hole is made in one of the walls of the square part containing the electrolyte. A glassy carbon plate (500 μm thickness) working electrode was fixed with glue from the outside in front of the hole in the electrochemical cell wall. The actual working electrode surface in contact with the electrolyte was ca. 0.5 cm$^2$ ($R = 0.4$ cm). A plastic disk was fitted in between the screw cap and the cell top entrance, with holes to allow other electrodes. The counter electrode was a Pt mesh inserted in a glass compartment in contact with the electrochemical solution through a fritted glass. The reference electrode was an

Ag/AgCl ($E = + 0.2$ V vs. NHE) electrode. The incoming photon intensity was monitored using a N$_2$/He-filled ionization chamber. In situ/operando XAS data were collected as fluorescence excitation spectra using a 36-pixels Ge detector (Canberra) with a 45° outgoing angle. Constant electrochemical potentials were applied to the working electrode for 5 min before the accumulation of XAS spectra. Each spectrum was accumulated in 10 min. for 20 min, during which time the potential was kept constant. The absorption edge E$_0$ was measured as the first inflection point in the derivative spectrum and calibrated vs. an Ir foil. The XANES and EXAFS spectra were normalized from 30 to 93 eV below the edge and 96 to 554 eV above the edge, respectively. For all XAS experiments (ex situ and in situ), uncertainties on energy values were estimated to be ca. ±0.25 eV.

## Data availability

The datasets generated during and/or analyzed during the current study are available from the corresponding author on reasonable request. Source data are provided with this paper.

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

## Acknowledgements

The work of M.O. was financially supported by College de France. This work was partially financed by SATT Lutech. M.F. acknowledges funding from the European Research Council (ERC) under European Union's Horizon 2020 Programme (Grant Agreement no. 803220, TEMPORE). The XAS experiments were performed on SAMBA beamline at Synchrotron Soleil (Proposal 20181713). We would like to acknowledge Guillaume Alizon for his assistance with the in situ XAS data measurements. The authors thank Maria de Marco for her help in spray-drying. Sophie Nowak is gratefully acknowledged for EDXRF measurements. Philippe Decorse is gratefully acknowledged for XPS experiments. XPS equipment was funded by the Région Île-de-France, convention SESAME n°16016303 and the Labex SEAM (Science and Engineering for Advanced Materials and devices). M.O. acknowledges Foundation for Polish Science (FNP) for a START scholarship.

## Author contributions

M.E., M.O., and M.Ga. performed material synthesis. S.D and C.T. performed electrochemical measurements and data analysis. M.B. and O.E. performed TEM experiments. J.-Y.P., C.B., C.S. contributed to materials characterization and fruitful discussion. M.G., A.Z., B.L.-K. conceived XAS experiments and analyzed the data. M.F. and J.P. supervised the project and wrote the manuscript.

## Competing interests

The authors declare no competing interests.

## Additional information

**Peer review information** *Nature Communications* thanks Elena Willinger and other, anonymous, reviewers for their contribtuions to the peer review of this work. Peer review reports are available.

