## [Peer Review File · Nature Communications]

REVIEWER COMMENTS

Reviewer #1 (Remarks to the Author):

Recommendation: Publish in Nature Communication.

Comments: The proposed manuscript describes an in-depth analysis of IrMo-mixed oxides for OER. By introducing molybdenum to IrO₂, the authors claim that they were able to distinguish the influence of the oxidation state from the structure of the material. The manuscript is well structured and written. The scientific content that builds on many experimental techniques clearly demonstrates the high level of understanding in the field. Findings and conclusions of the paper are important and well supported, which would in principle justify publication in Nature Communication.

The main conclusion of the paper is that the OER activity of the IrMo-mixed oxides does not depend so much on the Ir oxidation state, but rather on the structure of the material. From one side, they confirm earlier findings according to which the redox properties of the final structural motif, together with its structural flexibility, plays a key role for OER performance. From another side, in this manuscript, the authors very clearly and systematically demonstrate the importance of a local structural organisation of the material for the catalytic activity.

It would be nice if the authors would mention/consider in the manuscript a possible synergetic effect between Ir and Mo for OER performance. Probably valence band measurements of the samples with different Mo content and different annealing temperatures could shed more light on this point.

Elena Willinger

Reviewer #2 (Remarks to the Author):

This paper reports the aerosol synthesis method for the highly porous Ir-non-noble mixed metal oxides. The synthesized IrMo-mixed oxides present a unique behavior, which allows decoupling oxidation from crystallization processes occurring upon calcination. The authors explained that the high electrochemical activity of Ir-based oxides calcined at low temperature is unlikely due to the Ir oxidation state, but rather strongly depends on the material structure. However, many papers have shown that IrO_x, which has a high oxidation state, has high catalytic activity by using various in-situ/operando analysis. Therefore, the authors need to observe the oxidation state during real operating condition for OER, not ex-situ conditions. This is because even though the oxidation state is low under pristine conditions, there are many cases of showing a high oxidation state in the OER operating conditions. And this manuscript focuses too much on the synthesis of catalysts. Even though this paper shows an interesting concept for OER electrocatalyst, it is necessary to prove experimentally why the performance is improved. Considering the Nature Communications, which aims to explain theoretically or empirically, those reporting new understandings, the authors have not been able to provide sufficient evidence for improved catalytic activity. Hence, this manuscript is not acceptable for publication Nature Communications.

Reviewer #3 (Remarks to the Author):

The authors try to disentangle electronic and structural effects on the OER activity of amorphous Ir-oxides by studying IrMo oxides. The major claim is that Ir oxidation state is not linked to the observed high activity. Instead, the authors claim atomic structure and short-range order control activity. The topic is important. If the authors can substantiate the claims better, this work would be very important too, but currently the complex approach adds more ambiguity than it clears up.

The authors main claim is that IrMo mixed oxides can be used to decouple structural and oxidation

state changes occurring during calcination. They assert an Ir(III)/Ir(IV) transition occurs at a fixed temperature regardless of calcination temperature and Mo content from 10-50%, while structural changes and OER activity depend on calcination temperature. However, ex situ XPS and XAS appears to show the Ir electronic structure, oxidation state, changes during calcination up to the highest temperatures, depending on Mo content. The authors need to do a better job of quantifying these changes in a clear manner.

Mo 3d XPS also shows significant oxidation state changes associated with Mo, which will change the electronic structure of Ir. In fact, Mo is present as Mo(VI) and Mo(V) though the authors claim this Mo is in a single phase rutile-type IrO₂ structure. The Mo(VI) and Mo(V) will influence the electronic structure of Ir (see previous work on mixed IrMo in the OER DOI: 10.1021/acssuschemeng.7b04266). From this the authors cannot exclude changes in Ir oxidation state.

The exclusive use of ex situ XPS and XAS to assign oxidation states in the as-prepared materials is also dubious because the catalysts change under reaction conditions. For instance, in the introduction the authors cite Strasser's group proposing high OH concentration and Ir(III) as being linked to high activity. Such claims were generally from ex situ measurements. Pfeifer used in situ measurements to attribute such OH to the pre-catalytic state (DOI: 10.1039/C6SC04622C). For the authors' materials such transformations during operation may be more severe. The Pourbaix diagram of Mo shows MoO₃ should form under OER conditions. At a minimum the authors should perform post-reaction XPS, XRD, and SEM to help demonstrate changes in electronic structure cannot account for changes in activity.

In terms of activity, the authors claim Mo does not change the intrinsic activity of Ir based on the mass normalized currents. This is not convincing. Ir mass based activity is a cost parameter. The authors would need to know the number of Ir atoms accessible to the electrolyte to make their claim. The authors should give a measure of surface areas. The authors should also show steady-state data and perform a Tafel analysis to give a better picture of the activity. The authors may also need to test for mass transport limitations using different rotation speeds.

More detailed comments are given below:

XPS spectra should be fit and the quantification given.

An easier comparison with well calcined IrO₂ should be made in XAS. The authors show the first derivative peak shifts between 450 and 500 C, but the L3 white line intensities should also be analyzed to assign the average oxidation state.

EXAFS at the L3 edge is not ideal. A sufficiently large data range should include L2, which will interfere. The Ir K edge is more robust.

Replicate measurements would be done to test electrochemical performance.

Many figure axes and labels are illegible. The pdf conversion might have failed.

MoO₃ should be included in Supplementary Figure 2.

Answers to reviewers' comments.

Reviewer #1 (Remarks to the Author):

Recommendation: Publish in Nature Communication.

Comments: The proposed manuscript describes an in-depth analysis of IrMo-mixed oxides for OER. By introducing molybdenum to IrO₂, the authors claim that they were able to distinguish the influence of the oxidation state from the structure of the material. The manuscript is well structured and written. The scientific content that builds on many experimental techniques clearly demonstrates the high level of understanding in the field. Findings and conclusions of the paper are important and well supported, which would in principle justify publication in Nature Communication.

The main conclusion of the paper is that the OER activity of the IrMo-mixed oxides does not depend so much on the Ir oxidation state, but rather on the structure of the material. From one side, they confirm earlier findings according to which the redox properties of the final structural motif, together with its structural flexibility, plays a key role for OER performance. From another side, in this manuscript, the authors very clearly and systematically demonstrate the importance of a local structural organisation of the material for the catalytic activity.

It would be nice if the authors would mention/consider in the manuscript a possible synergetic effect between Ir and Mo for OER performance. Probably valence band measurements of the samples with different Mo content and different annealing temperatures could shed more light on this point.

Elena Willinger

We would like to thank the reviewer for her careful reading of the manuscript and her careful comments. We indeed believe that structure is the key-parameter to reach high activity. We also show that the activity is independent of the iridium oxidation state. Regarding the influence of Mo onto Ir, we have replotted Ir XPS spectra to have a better view of a possible chemical shift of Ir 4f signal upon variation of Mo content (Supporting Figure 6) and also provide a deeper analysis of XANES spectra to plot systematically edges and peak-top values (Supporting Figure 9). Even if these additional data do not completely rule out a possible effect of Mo on the electroactivity of the material, we do not observe any change in Ir XPS or XAS spectra when varying Mo content. We believe that this rule out a strong influence of Mo, result which is now clearly mentioned in the manuscript.

In addition to the initial work, we have also performed operando XAS experiments. They actually show that at 1.7 V vs. NHE, under OER conditions, the oxidation state of all the materials increases towards a similar value ($> \text{Ir(IV)}$). This very high oxidation state (formally $>+4$) observed at high potential is in agreement with other results from the literature. This observation confirms that the electroactivity of the material is independent of the initial oxidation state of the material. Strongly supporting our initial findings, we have decided to add these results in the main paper as highlighted in the revised version of the manuscript.

Reviewer #2 (Remarks to the Author):

This paper reports the aerosol synthesis method for the highly porous Ir-non-noble mixed metal oxides. The synthesized IrMo-mixed oxides present a unique behavior, which allows decoupling oxidation from crystallization processes occurring upon calcination. The authors explained that the high electrochemical activity of Ir-based oxides calcined at low temperature is unlikely due to the Ir oxidation state, but rather strongly depends on the material structure. However, many papers have shown that IrO_x, which has a high oxidation state, has high catalytic activity by using various in-situ/operando analysis. Therefore, the authors need to observe the oxidation state during real operating condition for OER, not ex-situ conditions. This is because even though the oxidation state is

low under pristine conditions, there are many cases of showing a high oxidation state in the OER operating conditions.

We would like to thank the reviewer for reading the manuscript and her/his comments. We agree that the oxidation state has been shown to increase during OER. We thus understand the reviewer's concern on the pertinence of using the initial oxidation state (ex-situ) as a parameter to investigate the electrochemical activity. This needs indeed to be clarified.

Several studies have been recently reported with the objective to unravel the mechanisms of OER by using operando techniques (10.1021/acs.jpcclett.8b00810, 10.1021/acscatal.5b01281, 10.1038/s41929-018-0153-y, doi.org/10.1039/C4SC00975D). In this paper, we propose principally to identify the key-parameters, in particular structure/oxidation state of Ir, for producing highly active catalysts, and that is why our work was mainly used ex-situ characterizations of the materials.

As requested by the reviewer we performed in-situ XAS measurements at synchrotron in OER conditions for monitoring the evolution of these parameters during operando measurements. As expected, these analyses show a higher oxidation state during OER for all the studied catalysts, confirming thus our initial statements and, by this means, strengthening our claims. In these experiments we monitored the evolution of oxidation state before, during and after electrochemical reaction by recording XANES spectra for a series of samples prepared with 30% Mo and calcined at different temperatures (450°, 500°C, 550°C). In the series, all the materials have the same composition but different initial oxidation state. The results are displayed in the graph thereafter.

XANES spectra of the samples prepared with 30% Mo and calcined at 450, 500 and 550 °C. Spectra were recorded in operando conditions prior measurement (0.2 V vs. NHE), at 1.7 V vs. NHE and back to 0.2 V vs. NHE. The values of peak top positions measured on the XAS spectra are reported on the bar diagram.

The initial XANES edge positions measured in the electrochemical cell at 0.2 V vs. NHE confirmed the results obtained from ex-situ measurements: the oxidation state is lower (Ir(III)+Ir(IV)) for materials calcined at 450°C, and then increases to Ir(IV) for materials calcined at > 500 °C.

At 1.7 V vs. NHE, under OER conditions, the oxidation state of all the materials increases and, all of them present similar oxidation state during OER (> Ir(IV)). This very high oxidation state (formally >+4) observed at high potential is in agreement with other results from the literature. This observation confirms that the electroactivity of the material is independent on the initial oxidation state of the material.

Coming back to 0.2 V vs. NHE, all the materials recover their initial oxidation state. A slight increase in the intensity of the white line for the material calcined at 450 °C can be observed, but the peak top position remains very similar to the initial one (11218.7 eV vs. 11218.9 eV). This difference probably arises from the conversion of residual Ir-Cl bonds into Ir-O bonds, but Ir remains in the +III oxidation state.

These results confirm one of our main claims which is that the electrochemical activity of the material is independent on the initial oxidation of the material. This is also strengthening our conclusions on the predominant role of the structure to explain the trend of the electrochemical activity of the samples since all the materials present similar oxidation state in OER conditions. In addition, this gives valuable information for preparing highly active electrocatalysts.

These results are now included in the main paper as additional figures and text and highlighted in the revised version of the manuscript.

Thanks to these results, we confirm that whatever the initial oxidation state in the material, the performance are optimal if the material is made of nanocrystallites of ca. 1.6 nm. However, for a similar oxidation state, the activity strongly depends on the crystalline state of the material.

And this manuscript focuses too much on the synthesis of catalysts. Even though this paper shows an interesting concept for OER electrocatalyst, it is necessary to prove experimentally why the performance is improved. Considering the Nature Communications, which aims to explain theoretically or empirically, those reporting new understandings, the authors have not been able to provide sufficient evidence for improved catalytic activity. Hence, this manuscript is not acceptable for publication Nature Communications.

In this paper, we do not claim that our materials exhibit improved performances. We highlight this point in Figure 2 where we show that the activity reported to the mass of iridium is independent on Mo content. Instead, by introducing molybdenum into the structure, we now are able to tune the catalyst's crystallinity and oxidation state. This allows us to solve a very important question regarding the high activity of poorly crystallized iridium oxides as mentioned above. We claim that for each serie of Mo-containing Ir oxides, the most active materials are those with characterized a typical short-range order corresponding to crystal size of around 1.6 nm. We believe that, to support our conclusions, we used a wide range experimental techniques including electrochemical measurement, XPS, XAS, XRD, electron microscopy for 27 different catalysts. We also believe that the in-situ experiments performed in synchrotron and inserted in the revised version as a new figure strengthen these findings.

Reviewer #3 (Remarks to the Author):

The authors try to disentangle electronic and structural effects on the OER activity of amorphous Ir-oxides by studying IrMo oxides. The major claim is that Ir oxidation state is not linked to the observed high activity. Instead, the authors claim atomic structure and short-range order control activity. The topic is important. If the authors can substantiate the claims better, this work would be very important too, but currently the complex approach adds more ambiguity than it clears up.

The authors main claim is that IrMo mixed oxides can be used to decouple structural and oxidation state changes occurring during calcination. They assert an Ir(III)/Ir(IV) transition occurs at a fixed temperature regardless of calcination temperature and Mo content from 10-50%, while structural changes and OER activity depend on calcination temperature. However, ex situ XPS and XAS appears to show the Ir electronic structure, oxidation state, changes during calcination up to the highest temperatures, depending on Mo content. The authors need to do a better job of quantifying these changes in a clear manner.

(A) - We would like to thank the reviewer for careful reading of the manuscript and her/his comments. As mentioned thereafter, we have replotted XPS spectra (Supplementary Figure 6), added figures to report more clearly edge shifts determined from XANES spectra (Supplementary Figure 9), and

performed additional experiments: operando XAS experiments. We believe that these additional data analysis and experiments strongly strengthen the paper thanks to the reviewer comments. Detailed answer are given below.

Mo 3d XPS also shows significant oxidation state changes associated with Mo, which will change the electronic structure of Ir. In fact, Mo is present as Mo(VI) and Mo(V) though the authors claim this Mo is in a single phase rutile-type IrO₂ structure. The Mo(VI) and Mo(V) will influence the electronic structure of Ir (see previous work on mixed IrMo in the OER DOI: 10.1021/acssuschemeng.7b04266). From this the authors cannot exclude changes in Ir oxidation state.

(B) - As outlined by the reviewer, from XRD we indeed observe a single phase corresponding to IrO₂-rutile type structure and we do not observe the presence of Mo-oxides. Using chemical analysis from TEM/EDX, we observe homogenous distribution of Ir and Mo throughout the material and therefore clearly show that we have formed a true mixed oxide and not a composite material.

To answer to the point on electronic effects, we have now performed deeper analysis of XAS and XPS spectra as reported below.

It is first important to recall that from XPS measurements, Mo(VI) is indeed observed and it is reduced into Mo(V) between 400 and 500 °C. This transformation, Mo(VI) into Mo(V), occurs in the same temperature range as the formation of the mixed oxide and the oxidation of Ir(III) into Ir(IV). In this region, it is therefore difficult to distinguish the oxidation process from electronic effects of Mo onto Ir if any. It then remains in the same oxidation state up to 600 °C. At 800 °C, a small contribution of Mo(IV) is observed along with Mo(V) (Figure S7) in all cases the samples calcined at 800 °C show very low performance (including pure iridium oxide).

To evidence the electronic influence of Mo onto Ir, we have plotted the position of the edge and the peak top position (commonly referred as white line) determined on iridium XANES spectra of each compound calcined at each temperature. On these two plots, we do not observe a particular trend of the top peak position or of edge value when Mo content in the material increases. Whatever the composition, for calcination temperature higher than 500 °C, *i.e.* when there is no influence of the oxidation process, the edge and white positions ranges from 11217.5eV and 11218 eV and 11219.8 and 11220.1 eV, respectively. Even if it is more difficult to draw affirmative conclusions from the samples calcined between 400 and 500 °C due to the concomitant oxidation, similar observation is made in this temperature range and the relative position of the points do not seem to depend on Mo content. This rule out the existence of a strong influence of Mo on Ir electronic structure. However, we clearly see that the edge and peak top positions increase of *ca.* 1 eV when the calcination temperature increases from 400 to 500 °C which indicates the transformation of Ir(III) into Ir(IV) species upon heating.

Figure B-1: (a) Peak-top and (b) edge values determined from XANES spectra at the Ir L₃-edge for samples prepared with 10 % Mo, 30 % Mo and 50 % Mo and calcined from 400 to 800°C, and IrO₂ calcined at 800 °C.

Similar conclusions can be made from the analysis of iridium XPS spectra. A representative example is depicted thereafter where we do not observe a particular trend of the Ir 4f5/2 and 4f7/2 peak position with Mo content variation (including when comparing with pure IrO₂) for samples calcined at 550 °C. (However, we can clearly see that the transition between Ir(III) and Ir(IV) occurs between 400 and 500 °C whatever the composition – see below answer (F)).

Figure B-2: XPS spectra of samples prepared with 10, 30 and 50% Mo and pure IrO₂ and calcined at 550 °C.

This is not in contradiction with the work reported in [10.1021/acssuschemeng.7b04266](https://doi.org/10.1021/acssuschemeng.7b04266) since in their case, the material is not a mixed oxide but a composite material made of MoO₃ large platelets and small IrO₂ nanoparticles deposited on the large platelets. However, we should mention that, even if the work reported in [DOI: 10.1021/acssuschemeng.7b04266](https://doi.org/10.1021/acssuschemeng.7b04266) is of high interest, we do not agree with the attribution of iridium peaks the authors did in this paper. Indeed, they state that Ir(IV) is located at 62.15 eV and 65.14 while this shift is usually attributed to Ir(III) including in the references cited in this paper ([DOI:10.1021/acsami.5b10159](https://doi.org/10.1021/acsami.5b10159), [DOI:10.1002/sia.5895](https://doi.org/10.1002/sia.5895), [DOI: 10.1039/C5CP06997A](https://doi.org/10.1039/C5CP06997A)). The binding energies of Ir(4f7/2) and 4f5/2) in IrO₂ single crystal values are 61.7 and 64.7 eV, respectively (in our work, they range between 61.58 and 61.78 and 64.58 and 64.78 respectively for samples calcined at T > 600 °C, whatever the Mo content). To our understanding, their material probably contains Ir(III) and the shift from 62.44 to 62.15 eV for the Ir(4f7/2) and from 65.37 to 65.14 eV for the Ir(4f5/2) cannot be unambiguously attributed to an electronic effect or to the presence of both Ir(III) and Ir(IV) species in different proportions.

We fully agree with the reviewer regarding the fact that on the figures of the previous version of the paper, it was indeed not possible to clearly see if Mo had an influence on the electronic structure of Ir. We have now added into Supplementary Figure 9 the two graphs mentioned above displaying the position of the edge and the peak top positions determined as a function of calcination temperature and Mo content from iridium XANES spectra recorded on pellets, and replotted Ir XPS spectra (see answer F below and Supplementary Figure 6).

The exclusive use of ex situ XPS and XAS to assign oxidation states in the as-prepared materials is also dubious because the catalysts change under reaction conditions. For instance, in the introduction the authors cite Strasser's group proposing high OH concentration and Ir(III) as being linked to high activity. Such claims were generally from ex situ measurements. Pfeifer used in situ measurements to attribute such OH to the pre-catalytic state ([DOI: 10.1039/C6SC04622C](https://doi.org/10.1039/C6SC04622C)). For the authors' materials such transformations during operation may be more severe. The Pourbaix diagram of Mo shows MoO₃ should form under OER conditions. At a minimum the authors should perform post-reaction XPS, XRD, and SEM to help demonstrate changes in electronic structure cannot account for changes in activity.

(C) - We agree that the oxidation state has been shown to increase during OER. We thus understand the reviewer's concerns on the utilization of the initial oxidation state (*ex-situ*) and on the possible structural transformation of Ir-based catalysts during OER.

To clarify these aspects, we have performed in-situ XAS measurements at synchrotron in OER conditions and observed the evolution of both the oxidation state and the material structure during operando measurements. In these experiments we monitored the evolution of oxidation state before, during and after electrochemical reaction by recording XANES spectra for catalysts prepared with 30% Mo and calcined at different temperatures (450°, 500°C, 550°C). This series of samples was chosen because all of them have the same composition but different initial oxidation state.

The results are shown thereafter.

Figure C-1: XANES spectra of the samples prepared with 30% Mo and calcined at 450, 500 and 550 °C. Spectra were recorded operando prior measurement (0.2 V vs. NHE), at 1.7 V vs. NHE and back to 0.2 V vs. NHE. The values of peak top positions measured on the XAS spectra are reported on the bar diagram.

The initial XANES peak top positions measured in the electrochemical cell at 0.2 V vs. NHE confirmed ex-situ measurements: the oxidation state is lower (Ir(III)+Ir(IV)) for materials calcined at 450°C and increases Ir(IV) for materials calcined at > 500 °C.

At 1.7 V vs. NHE, under OER conditions, as expected, the oxidation state of all the materials increases. More importantly, whatever the initial oxidation state, all of them tends to have a similar oxidation state during OER (> Ir(IV)). This very high oxidation state (formally >+4) observed at high potential is in agreement with results from the literature. This observation confirms our initial statement, i.e. that the electroactivity of the material is independent on the initial oxidation state of the material.

Coming back to 0.2 V, all the materials recover their initial oxidation state, which means that OER conditions did not affect the electronic structure of the material. There is a slight increase in the intensity of the white line for the material calcined at 450 °C but the peak top position remains very similar to the initial one (11218.7 eV vs. 11218.9 eV). This difference probably arises from the conversion of residual Ir-Cl bonds into Ir-O bonds, but Ir remains in the +III oxidation state.

These results confirm one of our main claims which is that the electrochemical activity of the material is independent on the initial oxidation of the material. This is also strengthening our conclusions on the predominant role of the structure to explain the trend of the electrochemical activity of the samples since all the materials presents similar oxidation state in OER conditions.

To assess the material structure evolution during OER, we have analyzed the Ir EXAFS spectra of the material before, during OER and after OER. The spectra of the materials prepared with 30% Mo and calcined at 500 °C recorded before, during and after OER are represented in Figure C-2. We can see that the spectra recorded before and after OER perfectly overlap. This rules out a strong/irreversible evolution of the material structure (Ir local range order) in OER conditions in the course of the experiment.

Figure C-2: EXAFS spectra of the most electrochemically active materials of the 30% Mo series (calcined at 500 °C). Spectra were recorded operando following the sequence: 0.2 V vs. NHE (black diamonds), at 1.7 V vs. NHE (pink circles) and back to 0.2 V vs. NHE (blue triangles).

Regarding further analysis by XRD, SEM or XPS after in-situ experiment, due to the very low loading of the material in the electrochemical cell and the mixing with Nafion and carbon for the electrode preparation, the measurements would be strongly affected. However we believe that in-situ XAS experiments and XANES and EXAFS spectra analysis, showing no evolution of the oxidation state or of the material structure between initial and post-OER experiment are sufficient to demonstrate that changes in electronic structure cannot account for changes in activity.

Moreover, we believe that, thanks to the reviewer suggestions, these additional experiments strongly strengthen our claims: i.e. the activity is related to the material structure but does not depend on the initial oxidation state. In addition, this gives valuable information for preparing highly active electrocatalysts. For these reasons, we have decided to include these results in the main paper as additional figures and text which are highlighted in the revised version of the manuscript.

In terms of activity, the authors claim Mo does not change the intrinsic activity of Ir based on the mass normalized currents. This is not convincing. Ir mass based activity is a cost parameter. The authors would need to know the number of Ir atoms accessible to the electrolyte to make their claim. The authors should give a measure of surface areas.

(D) - We understand the reviewer's concern on the mass normalized current. However the accurate determination of the electrochemical active surface area or number of participating atoms in the catalytic materials is a real challenge in the case of iridium oxides since IrO₂ cannot undergo proton underpotential deposition like platinum and other transition metal electrocatalysts. (10.1149/2.0211512jes) This is even more challenging for mixed iridium oxides including another transition metal like Mo. The determination of surface area by physisorption gives an idea but does not reflect real electrochemical surface area and going further into the exact number of iridium atoms would however be too dubious. For these reasons, even if not ideal, we preferred to indicate the mass normalized current that can be determined in a reliable way. We totally understand that this activity is not proportional to the number of active iridium atoms, however, since the activity is normalized by the mass of iridium and not by the mass of oxide, it is at least directly proportional to the total number of iridium atoms. This normalization method is also employed in several papers dealing with IrO_x particles (e.g. 10.1021/acs.chemmater.6b02625, 10.1002/adfm.201906670), unlike for thin films where it is usually normalized by the geometrical surface of the electrode.

The main reviewer's question is related to possible change in Ir activity due to Mo. To further clarify this aspect, as explained in answers (B) and (C), we also provide more detailed analysis of XPS, ex-situ

XAS experiments and additional in-situ XAS experiments. These data indicates that the Mo presence does not affect significantly the electronic structure of Ir before OER and during OER. We believe that, these new results provide an additional indication that the presence of Mo is not playing a predominant role on the Ir activity.

The authors should also show steady-state data and perform a Tafel analysis to give a better picture of the activity. The authors may also need to test for mass transport limitations using different rotation speeds.

(E) - We have included the Tafel plots and reported the Tafel slope for all the samples in a Supplementary Figure (SI5) as follow:

Figure E-1: Tafel plots represented for the 50th forward scan and calculated corresponding Tafel slopes of samples obtained after heat treatment in air at 400 °C, 450 °C, 500 °C, 550 °C and 600 °C. For most of samples, the Tafel slopes range from 70 and 90 mV/dec with a typical value of 70-74 for the most active materials of each series. The highest Tafel slopes are reported for fully amorphous materials”.

Regarding the mass transport limitations, the very high porosity of the particles facilitate gases and water transport throughout the materials and, with the loading we have used for electrochemical characterization, we never observed instability (fluctuations at high current density) which could impede the measurements. In addition, each measurements were repeated at four times and error bars show that the measurements are highly reproducible. (Error bars were actually included in the initial figure, however there are so small that we cannot properly see them on the figure. We have now replotted the initial graph to be able to visualize the error bars – see answer I).

More detailed comments are given below:

XPS spectra should be fit and the quantification given.

(F) - As an electron conductor, IrO₂ has an asymmetric core level spectra and for this reason, highly reliable quantification of the Ir oxidized species (Ir(IV), Ir(III)) using XPS is challenging, and caution is required for the speciation of iridium oxides as highlighted in papers dedicated to the electronic structure of iridium oxides (10.1002/sia.5895 , 10.1039/C5CP0699). We therefore think that a precise deconvolution of iridium 4f region would be the source of too high uncertainty. However, we fully agree with the reviewer that the iridium spectra shown in the previous version of the paper were not sufficiently clear to support our findings. We have therefore re-plotted all the spectra presented in Supporting Figure 6, and the new graphs give now a better view of the evolution of the Ir 4f_{7/2} and Ir 4f_{5/2} components as a function of the calcination temperature, and this for the 3 samples made with 10, 30 or 50% of Mo. We also provide detailed explanation of the spectra as detailed thereafter.

Figure F-1: XPS spectra in the Ir 4f region of samples prepared with (a) 10% Mo, (b) 30% Mo and (c) 50% Mo and calcined from 400 to 800 °C.

“Extensive studies of iridium oxidation states by XPS have been reported by Pfeifer et al. Coupling experimental with theoretical studies, these authors proposed that Ir(III) centres exhibit a reverse binding shift as already observed for Ag.[10.1002/sia.5895, doi.org/10.1039/C5CP06997A, doi.org/10.1039/C6SC01860B] Ir(III) species usually show two peaks centered at 62.3 to 62.5 eV and 65.3 to 65.5 eV. As an electron conductor, IrO₂ has an asymmetric core level spectra, and asymmetric peaks characteristics of Ir(IV) in IrO₂ usually point at 61.7 to 61.9 eV and 64.7 to 64.9 eV for the 4f_{7/2} and 4f_{5/2} components, respectively. In our case, for the three samples, the XPS spectra of the powder calcined at 400 °C show two broad peaks centered at 62.3 and 65.3 eV that can be attributed to the presence of iridium (III) either resulting from residual chloro-iridium complexes or hydroxylated iridium species probably partially hydrated during the synthesis process. For samples calcined at 500 °C and above, all the spectra overlap and show two sharp peaks centered at 61.8 eV and 64.8 eV that can be attributed to the 4f_{7/2} and 4f_{5/2} components of the Ir(IV) in IrO₂ or Ir_xMo_{1-x}O₂. The oxidation of iridium species at the surface of the particles is already complete at 500 °C whatever the composition. For samples calcined at 450 °C, we observe an intermediate situation and samples obviously contain both Ir(III) and Ir(IV) species.”

An easier comparison with well calcined IrO₂ should be made in XAS. The authors show the first derivative peak shifts between 450 and 500 C, but the L3 white line intensities should also be analyzed to assign the average oxidation state.

(G) - As shown in Figure B-1 above, we have plotted the position of the edge and the peak top position (commonly referred as white line) determined on iridium XANES spectra of each compound calcined at each temperature. The edge values of materials calcined at 500 °C and above are similar to that of IrO₂ calcined at 800 °C (*i.e.* Ir(IV)). The values of materials calcined at 400 °C are *ca.* 1eV lower which correspond to Ir(III). These results are now inserted in the paper in Supporting Figure 9.

EXAFS at the L3 edge is not ideal. A sufficiently large data range should include L2, which will interfere. The Ir K edge is more robust.

(H) - Indeed, for some transition element, studying K-edge gives valuable information, however, Ir K-edge is not suitable for studying the electronic structure and structural properties of such systems. Indeed, Ir K-edge is at 76111 eV. For absorption edges above 40 keV, the structural signal can be strongly damped. In fact, the higher the energy of the absorption edge, the shorter the lifetime of the excited atomic state, and the greater the damping and broadening of the signal. The core-hole lifetime adds a Lorentzian component (Γ) to the lineshape with a large width ($\Gamma=5.25$ eV at Ir L3-edge, while $\Gamma=46.8$ eV at Ir K-edge; *J. Phys. Chem. Ref. Data, Vol. 8, No.2, 1979*) and it induces a number of problems: 1) a serious broadening of the XAS spectrum, then blunt edge jumps and reduction of EXAFS amplitude; 2) measurement are carried out with poor energy resolution; 3) the low energy resolution makes it difficult to evaluate the correct EXAFS signal especially at low-k values; 4) the high-frequency components of the measured signal are more affected, and consequently this reduces the sensitivity of the technique to the more distant coordination shells. An example of these undesirable effects is illustrated in the reference *J. Synchrotron Rad. (1998). 5, 1007-1009*.

Replicate measurements would be done to test electrochemical performance.

(I) - Error bars were actually included in the initial figure coming from 4 independent measurement for each sample, however there are so small that we couldn't properly see them. We have decreased the marker sizes and increased error bar marker size. We have also plotted the error bars in black (different color from the marker) and it is now possible to view them in most cases as shown in Figure I-1. We have also modified the legend to specify that error bars correspond to the standard deviation between measurements performed on three different deposited inks.

Figure I-1: Electrochemical activity of the materials. (a) Current densities values recorded at 1.5V vs. NHE for pure IrO₂ (diamonds, black) samples prepared with 10% (crosses, blue), 30% (squares, orange) and 50% (circles, green) of Mo as a function of temperature of calcination – lines are guide for the eyes. SD error bars calculated from measurements performed on four different deposited inks are included

and represented in black for each point; (b) Current density recorded at 1.5 V vs. NHE for each sample at the optimal calcination temperature. The current density is reported to the geometrical surface of the electrode (plain bars) or to the mass of iridium calculated using the stoichiometry of the mixed oxide (dashed bars). SD error bars included.

Many figure axes and labels are illegible. The pdf conversion might have failed.

(J) – We have re-printed all the figures. It looks ok to us.

MoO₃ should be included in Supplementary Figure 2.

(K) - Done.

REVIEWER COMMENTS

Reviewer #2 (Remarks to the Author):

The authors have performed well in-situ/operando XAS measurements in OER conditions and this will be a good result for observing the actual behavior of the prepared catalysts. However, there are a few more questions about these results. The manuscript will be allowed to be published in Nature Communications once these points have been confirmed. Some of the questions about this manuscript are following:

1. The authors pointed out that the oxidation state of Ir does not play an important role in the OER catalytic activity. However, this is a wrong conclusion. Since the particle size and crystallinity are different for each catalyst, the average oxidation state measured up to the bulk phase and the oxidation state of the surface where OER reaction occurs may be different. The prepared IrMo oxide is synthesized by a heat treatment method and has high crystallinity in the core region, and even if an amorphous Ir oxide with a high oxidation state is formed on the surface of the catalyst during OER, it can have a low oxidation state in XAFS, which measures bulk properties. In order to mention that OER catalytic activity has no effect on the Ir oxidation state, it is necessary to measure the change in the electronic structure of the surface during the OER through a soft X-ray based in-situ/operando analysis techniques such as XPS and NEXAFS. Because the initial oxidation state of the catalyst does not affect the oxidation state during the OER, ex-situ XPS results cannot explain the effect of OER catalytic activity and oxidation state. Otherwise, it should be deleted that the OER catalytic activity is not affected by Ir oxidation state.

2. Many manuscripts reported that amorphous IrOx possess better performance than crystalline IrOx. However, author suggests that 1.6 nm crystalline Iridium oxide catalysts showed optimal performance than amorphous IrOx and bigger crystalline IrO₂. Generally, smaller crystalline size catalyst exhibits higher catalytic activity owing to high active surface area. Thus, this is not major novelty of this manuscript. Wasn't the improved OER catalytic activity of the prepared catalyst simply due to the structural features resulting from the small particle size and large active surface area?

3. The key part of this manuscript is the comparison of amorphous IrMo catalyst and 1.6 nm Ir MoO₂ catalysts. However, amorphous IrMo catalyst in this manuscript is not reported amorphous IrOx. Reported amorphous IrOx catalyst exhibited outstanding performance is typically synthesized by electrochemically activation. This showed higher oxidation state (XANES) than crystalline IrOx. However, amorphous IrMo catalyst in this manuscript showed lower oxidation state and chemical composition of IrMoOxCl_y which is different to IrO₂, showing poorly made IrO₂. Thus, author compares an incompletely made amorphous IrMoO₂ and completely made IrMoO₂. I think that this manuscript cannot provide new insight for amorphous iridium oxide and crystalline IrOx.

4. If you want to compare the performance of an electrochemically made amorphous oxide, activation process (0 ~ 1.5 VRHE, 500 mV s⁻², 100 cycles) was suggested. After activation process, amorphous catalyst surface would convert to electrochemically oxidized amorphous IrOx. However, crystalline IrO₂ almost maintains their state. The OER performance comparison of synthesized catalyst before and after activation process can offer information for debate of amorphous IrOx and crystalline IrO₂.

Reviewer #3 (Remarks to the Author):

The revised version of the manuscript is improved. The authors have fixed a number of small errors and even included an in situ XAS measurement set. However, the data still does not support the conclusions.

The authors continue to claim "an optimal degree of ordering" can explain the activity of X-ray

amorphous IrO_x and rule out the accepted picture of iridium oxidation state as a factor. The data used for this claim remains inconclusive for a variety of reasons:

The samples show varying Cl content, a poison.

It is not clear how the active surface area is changing between samples.

Mo oxidation state changes and correlates with activity.

It is not clear what the ratio of Mo/Ir on the surface is or if it is changing.

Mo introduces variations in the lattice parameters.

It is not clear if different facets are exposed upon alloying/annealing.

All of these could be influencing activity. It has been established that, for instance, surface termination/average surface oxidation state influences activity, see ACS Catal. 2020, 10, 806–817 and J. Phys. Chem. Lett. 2014, 5, 1636–1641. And while such variations are quantifiable, it is not clear what "an optimal degree of ordering" is. The authors should quantify ordering and show the variations expected to influence activity do not, e.g. area, faceting, oxidation state, strain.

The authors still use principally ex situ characterization techniques, and there is value in correlating ex situ results with activity. But the authors present no evidence any such correlation is anything but accidental. The in situ XAS, while potentially useful, is not analyzed in depth and was only performed for one sample. As the authors note, this technique is bulk sensitive and, for the large particles used in this work, will show only minor changes due to the expected differences in surface oxidation during operation.

If the authors are able to show ordering mediates rate, an atomic scale mechanism would be helpful. That is, how does ordering change the rate?

I also found the discussion of the material confusing. The authors claim to have Ir_{0.5}Mo_{0.5}O₂. From XPS Mo is Mo(VI) and Mo(V) and Ir is Ir(IV). This cannot be Ir_{0.5}Mo_{0.5}O₂. The authors should clarify this issue.

Answers to reviewers' comments.

Reviewer #2

The authors have performed well in-situ/operando XAS measurements in OER conditions and this will be a good result for observing the actual behavior of the prepared catalysts. However, there are a few more questions about these results. The manuscript will be allowed to be published in Nature Communications once these points have been confirmed. Some of the questions about this manuscript are following:

We would like to thank the reviewer for careful reading of the manuscript and her/his comments. Our answers are provided thereafter.

1. The authors pointed out that the oxidation state of Ir does not play an important role in the OER catalytic activity. However, this is a wrong conclusion. Since the particle size and crystallinity are different for each catalyst, the average oxidation state measured up to the bulk phase and the oxidation state of the surface where OER reaction occurs may be different. The prepared IrMo oxide is synthesized by a heat treatment method and has high crystallinity in the core region, and even if an amorphous Ir oxide with a high oxidation state is formed on the surface of the catalyst during OER, it can have a low oxidation state in XAFS, which measures bulk properties. In order to mention that OER catalytic activity has no effect on the Ir oxidation state, it is necessary to measure the change in the electronic structure of the surface during the OER through a soft X-ray based in-situ/operando analysis techniques such as XPS and NEXAFS. Because the initial oxidation state of the catalyst does not affect the oxidation state during the OER, ex-situ XPS results cannot explain the effect of OER catalytic activity and oxidation state. Otherwise, it should be deleted that the OER catalytic activity is not affected by Ir oxidation state.

Reply: Thank you for your comment. As suggested by the reviewer, after the first round of review, we had previously added *in situ* experiments. We fully agree with the reviewer with the fact that XAS is not appropriate to probe surface reactions occurring onto bulk materials. However, our materials have been designed to overcome these limitations and from the reviewer comment, we understand that we had not properly described the morphology of our materials and discussed this aspect in the previous versions of the manuscript.

We have now added a whole paragraph at the beginning of the paper and a new figure to fully describe the materials and explain why *in situ* and *ex situ* XAS measurements are relevant.

The new Figure is shown thereafter.

Figure 1: (a) Schematic illustration of the synthesis process for the formation of porous $\text{Ir}_x\text{Mo}_{(1-x)}\text{O}_y$ catalysts represented in the TEM micrograph (right side). SEM micrographs sequence of IrO_2 , $\text{Ir}_{0.9}\text{Mo}_{0.1}\text{O}_2$, $\text{Ir}_{0.7}\text{Mo}_{0.3}\text{O}_2$ and $\text{Ir}_{0.5}\text{Mo}_{0.5}\text{O}_2$ porous catalysts obtained at 500 °C. (c) Illustration of the morphology of a typical nanocrystalline catalyst. (d) and (f) HR-TEM micrographs of a single pore and of nanocrystalline wall respectively. (f) STEM image and corresponding EDX analysis of a catalysts containing 30% of Mo and calcined at 550 °C: (h) Mo L edge (green), (i) Ir M edge (red) and (g) overlapping of the two elements.

The morphology of a typical mixed-oxide catalyst is thus illustrated in Figure 1(c): the microspheres are highly porous (typically 75% vol.) composed of an interconnected macroporosity with thin walls that can be amorphous or composed by isotropic nanocrystals depending on the calcination temperature.

Each macropores has a size of ca. 250 nm. The **walls of the porous microsphere are very thin**, typically below 15 nm, as determined by HR-TEM. The pore's **walls are composed only of a few layers of crystalline nanoparticles** in the case of crystalline materials. In most cases, the particles are very small. The crystallite size were displayed in the main paper (currently Figure 4a), but for the sake of clarity, we are also reporting numerical values in Table S4.

Supporting Table 4: *Crystallite size determined by Retvield refinement of the corresponding X-ray diffractograms as a function of calcination temperature. Most active materials in each series is reported in bold character. *The crystallite size determined for pure IrO_x materials is an average between size calculated from a and from c parameters.(see ref. 10.1002/aenm.201802136 for details).*

Calcination temperature (°C)	Crystallite size (nm)			
	10% Mo	30% Mo	50% Mo	0% Mo
400	n.a.	n.a.	n.a.	n.a.
450	1.5	n.a.	n.a.	8.6*
500	6	2.5	n.a.	10*
550	7	4	2	12.5*
600	7.5	4.5	2.5	15.8*
800	11	7	5	27.7*

Therefore, even if the point raised by the reviewer about the accuracy of XAS in detecting the oxidation state in relation to the crystalline properties is very pertinent, the concept of bulk techniques becomes less relevant in the case of our materials. Indeed, the material was designed with this morphology because it is ideally suited for fundamental studies also by bulk sensitive techniques. The high accessibility of the porosity and the thin walls lead to a maximization of the Ir-based material involved in the catalytic reactions. This allow probing both surface and core Ir-based materials by XAS during operando experiments.

This can be confirmed experimentally by XAS. To clarify that, we have modified Figure 3: we have merged the discussion about ex situ and in situ XAS measurements, we have included additional SEM micrograph of the samples, and we have re-written the paragraph on iridium oxidation state. Part of this new Figure is presented thereafter.

series 30% Mo

Part of Figure 3: (b-d) SEM images of the series prepared with 30% Mo and calcined at 450 °C, 500 °C and 550 °C and (e-f) corresponding operando characterization at the Ir L3-edge: (e) XANES spectra recorded at 0.2 V vs. NHE, 1.7 V vs. NHE and back to 0.2 V vs. NHE and the corresponding peak-top values; (f) Superposition of the EXAFS spectra recorded for the most electrochemically active materials of the 30% Mo series (calcined at 500 °C) at 0.2 V vs. NHE (black diamonds), 1.7 V vs. NHE (pink circles) and back to 0.2 V vs. NHE (blue triangles).

The new SEM micrographs of the three samples displays that the materials have similar morphology, but while the sample calcined at 450 °C is amorphous, the samples calcined at 500 °C and 550 °C exhibit crystalline pore's walls. From these SEM micrographs, we believe we better highlight the peculiar porous morphology with thin pore's wall.

This allows for probing the iridium atoms located at the outer surface and in the inner porosity by *in situ* XAS experiments.

XANES spectra of the materials recorded at various potential and corresponding peak top values are plotted on Figure 3Part of Figure e. The initial XANES peak top positions measured in the electrochemical cell at 0.2 V vs. NHE confirmed *ex situ* measurements: the oxidation state is lower (corresponding to Ir(III)+Ir(IV)) for materials calcined at 450 °C and increases to Ir(IV) for materials calcined at 500 °C and 550 °C. At 1.7 V vs. NHE, under OER conditions, as expected, the oxidation state of all the materials increases. More importantly, whatever the initial oxidation state, all of them tends to have a similar oxidation state during OER (> Ir(IV)). This higher oxidation state (formally >+4) observed at high potential is in agreement with results from the literature where it has been shown that the Ir oxidation state can increase at high potential (10.1021/acs.jpcclett.8b00810, 0.1038/s41929-018-0153-y, 10.1039/C0CP01472A). Since the XAS experiment probes all the Ir atoms of the material, this significant increase in XANES peak top position (comparable with those reported by Strasser and co-workers in ref. 10.1038/s41929-018-0153-y) is an additional evidence that the reaction is not occurring only at to outer surface of the micrometric spheres but also it also inside the microspheres. This would not be the case if the core iridium materials were not accessible as in the case of not porous materials.

Indeed, in light of these results, we can confirm that whatever the initial oxidation state of our catalysts, during OER the Ir oxidation state increases toward similar values. This implies that the differences in electrochemical activity measured for the 30% Mo series (calcined at 450 °C, 500 °C and 550 °C) do not depend on the oxidation state during OER. In addition, from EXAFS spectra (Figure 3f) the reversibility of the process indicates that there is no major structural or chemical changes after OER. The experiments performed both *ex situ* and *in situ* indicate

that the electroactivity trend shown cannot be correlated with initial iridium oxidation state, nor the one measured *operando*.

Actions taken:

New Figure: Figure 1

Modification of Figure 3

New section: to better describe the formation/morphology of our materials

Modified section: possible influence of iridium oxidation state

New table: crystallite size, supporting Table 4

2. Many manuscripts reported that amorphous IrOx possess better performance than crystalline IrOx. However, author suggests that 1.6 nm crystalline Iridium oxide catalysts showed optimal performance than amorphous IrOx and bigger crystalline IrO₂. Generally, smaller crystalline size catalyst exhibits higher catalytic activity owing to high active surface area. Thus, this is not major novelty of this manuscript. Wasn't the improved OER catalytic activity of the prepared catalyst simply due to the structural features resulting from the small particle size and large active surface area?

Reply: Thank you for the remark. Indeed, the main conclusion of our study is that structural features drives electrochemical activities regardless the iridium oxidation number. We believe it is a major novelty in the field because usually, it was not possible to decorrelate the influence of structure from that of iridium electronic state. It is a key information for developing new highly active materials.

In this regards, the reviewer is right, improved activities corresponds to small crystallite size that also result in high surface area. To better investigate these aspects, we have now performed BET measurements to probe the evolution of the surface area of our materials and this is now discussed in the paper in a new section called: influence of surface area and crystallinity.

The value of surface areas, BET isotherms and crystallites size are reported in supporting Figure 15 and supporting Table 4, respectively.

Supplementary Figure 15: BET

In each series, we observe that the surface area increases up to the crystallization temperature. Indeed the beginning of the crystallization corresponds to formation of small nanoparticles characterized by a high surface

area. Increasing the calcination temperature leads to crystal growth that is usually accompanied by a decrease of surface area. This decrease is clearly observed for Ir and IrMo 10% characterized by a larger crystal sizes. However, the surface area of IrMo materials 30% and 50% does not vary and stabilize at ca. $70 \text{ m}^2 \text{ g}^{-1}$; this is due to the formation of mesopores within the walls which counterbalances the decrease in surface area that should be observed due to particle growth (as proved by the BET isotherms in Figure S15). On one hand, the increase of the activity between amorphous and crystalline materials could be attributed to an increase in surface area. On the other hand, the decrease in activity observed between very small crystallites and larger ones cannot be attributed only to the variation in surface area that remains almost constant.

This is not surprising since combining isotope labelling and atom probe tomography, Schweinar et al. (10.1021/acs.jpcclett.0c01258) have shown recently that during OER, even in the case of rutile IrO_2 , O atoms could be exchange between the oxide lattice and water down to 2.5 nm below the surface, and this depth could be even higher in the case of amorphous materials. If this is the case, this would means that for ultrasmall particles, all the atoms could participate in the catalytic reaction, making surface area a less relevant descriptor.

Actions taken:

New section: Influence of surface area and crystallinity.

New supporting figure: supporting Figure 15 on surface area measurements.

New supporting Table: supporting Table 4 for crystallite size.

3. The key part of this manuscript is the comparison of amorphous IrMo catalyst and 1.6 nm Ir MoO₂ catalysts. However, amorphous IrMo catalyst in this manuscript is not reported amorphous IrOx. Reported amorphous IrOx catalyst exhibited outstanding performance is typically synthesized by electrochemically activation. This showed higher oxidation state (XANES) than crystalline IrOx. However, amorphous IrMo catalyst in this manuscript showed lower oxidation state and chemical composition of IrMoOxCl_y which is different to IrO₂, showing poorly made IrO₂. Thus, author compares an incompletely made amorphous IrMoO₂ and completely made IrMoO₂. I think that this manuscript cannot provide new insight for amorphous iridium oxide and crystalline IrOx.

Reply: There are two important points that are raised by the reviewer here. First, the composition of our amorphous IrMo materials.

Indeed, we agree that during the thermal treatment we convert chlorinated precursors into mixed oxides but by playing on the Ir/Mo ratio, we can still obtain well-formed, amorphous materials with very low chlorine content as their crystalline counter part. To clarify this point, measured chlorine content by XPS for all the samples are reported, and, in addition we have now fitted EXAFS spectra of the samples prepared without Mo and with 30% and 50% Mo to determine the relative evolution of Ir-Cl and Ir-O with calcination. The average number of Cl and O in the coordination sphere of Ir from EXAFS spectra and the determined chlorine content as a function of the calcination temperature from XPS experiments are reported in Table S2 and Table S3, respectively.

Supporting Table 2a: Summary of the EXAFS fit Parameters for the pure IrO₂ calcined at different temperatures and recorded at the calcination temperature (except the 800 °C sample which was also recorded at RT.)^a

Sample	Shell	N	R / Å	$\sigma^2 / \text{Å}^2$	$\Delta E_0 / \text{eV}$	R _f (%)	χ^2_{red}
0% 400 °C	Ir-O	1.10 0.38	2.00 0.05	0.006 (fixed)	7.98 1.54	0.77	767.95
	Ir-Cl	4.44 0.33	2.35 0.01	0.005 (fixed)			
0% 450 °C	Ir-O	2.52 0.18	1.97 0.01	0.006 (fixed)	7.52 0.80	0.41	387.99
	Ir-Cl	3.08 0.16	2.34 0.01	0.006 (fixed)			
0% 500 °C	Ir-O	5.74 0.44	1.98 0.01	0.006 0.001	10.30 0.89	0.75	363.68
0% 550 °C	Ir-O	5.84 0.45	1.98 0.01	0.004 0.001	10.85 0.94	1.20	571.49
0% 650 °C	Ir-O	5.79 0.80	1.98 0.01	0.004 0.001	10.86 1.61	3.87	947.25

0% 800 °C	Ir-O	6 (fixed)	1.97 0.01	0.005 0.001	10.26 0.82	0.67	137.54
0% 800 °C (at RT) ^b	Ir-O	6 (fixed)	1.98 0.01 (DRX 2.00)	0.002 0.001	10.87 0.75	1.52	4442.11
	Ir-Ir ₁	2 (fixed)	3.14 0.01 (DRX 3.16)	0.002 0.001			
	Ir-O ₂	4 (fixed)	3.56 0.03 (DRX 3.41)	0.004 0.004			
	Ir-Ir ₂	8 (fixed)	3.54 0.01 (DRX 3.56)	0.001 0.001			

^a N, coordination numbers, r, distances between Ir and neighboring atoms, σ , Debye Waller factors, ΔE_0 , scaling energy parameter and R_f goodness of fit. Bold numbers correspond to fixed values and uncertainties are given in italic. Those parameters were obtained by least square fitting over a 1.0-3.0 Å range of the Fourier transforms of the experimental EXAFS spectra ($k_{\min} = 2.8 \text{ \AA}^{-1}$ and $k_{\max} = 10.6 \text{ \AA}^{-1}$, Hanning window with $dk = 1$). The reliability of the fit was assessed by the minimization of the reduced χ^2_{red} and metrics defined by the IXS standards and criteria committee (http://ixs.iit.edu/subcommittee_reports/sc/err-rep.pdf).

^b The fit of this sample was used to determine the value of the amplitude reduction factor $S_0^2 = 0.80$. The mean distances determined from the crystallographic structure for IrO₂ are reported for comparison purpose. Those parameters were obtained by least square fitting of over 1.1-4.0 Å of the Fourier transforms of the experimental EXAFS spectra ($k_{\min} = 3.0 \text{ \AA}^{-1}$ and $k_{\max} = 14.0 \text{ \AA}^{-1}$, Hanning window with $dk = 1$).

Supporting Table 2b: Summary of the EXAFS fit Parameters for the 30% Mo series calcined at different temperatures and recorded at RT.^a

Sample	Shell	N	R / Å	$\sigma^2 / \text{\AA}^2$	$\Delta E_0 / \text{eV}$	R_f (%)	χ^2_{red}
30% 400 °C	Ir-O	2.11 0.53	2.02 0.02	0.003 0.002	10.13 0.83	1.76	1129.68
	Ir-Cl	3.80 0.49	2.35 0.01	0.004 0.001			
30% 450 °C	Ir-O	2.85 0.86	2.01 0.02	0.002 0.002	10.27 1.82	12.34	1179.96
	Ir-Cl	2.05 1.02	2.35 0.02	0.004 0.004			
30% 500 °C	Ir-O	5.56 0.37	1.99 0.01	0.002 0.001	10.44 0.87	4.21	1127.15
30% 550 °C	Ir-O	5.79 0.29	1.98 0.01	0.003 0.001	10.26 0.67	1.85	894.59
30% 600 °C	Ir-O	5.98 0.30	1.98 0.01	0.003 0.001	10.03 0.65	1.60	2079.70
30% 800 °C	Ir-O	5.89 0.29	1.98 0.01	0.002 0.001	10.64 0.64	1.54	2232.25

^a N, coordination numbers, r, distances between Ir and neighboring atoms, σ , Debye Waller factors, ΔE_0 , scaling energy parameter and R_f goodness of fit. Bold numbers correspond to fixed values and uncertainties are given in italic. Those parameters were obtained by least square fitting over a 1.0-3.0 Å range of the Fourier transforms of the experimental EXAFS spectra ($k_{\min} = 2.8 \text{ \AA}^{-1}$ and $k_{\max} = 10.6 \text{ \AA}^{-1}$, Hanning window with $dk = 1$).

Supporting Table 2c: Summary of the EXAFS fit Parameters for the 50% Mo series calcined at different temperatures and recorded at RT.^a

Sample	Shell	N	R / Å	$\sigma^2 / \text{\AA}^2$	$\Delta E_0 / \text{eV}$	R_f (%)	χ^2_{red}
50% 400 °C	Ir-O	2.18 0.25	2.01 0.01	0.003 0.001	10.48 0.41	2.73	88.88
	Ir-Cl	3.73 0.21	2.35 0.01	0.004 0.001			
50% 450 °C	Ir-O	3.59 0.25	2.00 0.01	0.005 0.001	10.47 0.35	2.69	159.42
	Ir-Cl	2.66 0.18	2.35 0.01	0.003 0.001			
50% 500 °C (Oxygens only)	Ir-O	5.76 0.24	1.99 0.01	0.003 0.001	10.87 0.53	1.35	851.11
50% 500 °C	Ir-O	5.60 0.32	1.99 0.01	0.003 0.001	10.79 0.59	1.12	993.43

	Ir-Cl	0.15 0.21	2.34 0.05	0.001 0.001			
50% 550 °C	Ir-O	6.03 0.28	1.98 0.01	0.003 0.001	10.53 0.60	1.35	1453.51
50% 600 °C	Ir-O	5.76 0.31	1.99 0.01	0.002 0.001	10.94 0.70	1.52	1027.26
50% 800 °C	Ir-O	5.90 0.27	1.98 0.01	0.002 0.001	10.74 0.60	1.19	281.97

^a N, coordination numbers, r, distances between Ir and neighboring atoms, σ , Debye Waller factors, ΔE_0 , scaling energy parameter and R_f goodness of fit. Bold numbers correspond to fixed values and uncertainties are given in italic. Those parameters were obtained by least square fitting over a 1.0-3.0 Å range of the Fourier transforms of the experimental EXAFS spectra ($k_{\min} = 2.8 \text{ \AA}^{-1}$ and $k_{\max} = 10.6 \text{ \AA}^{-1}$, Hanning window with $dk = 1$).

Supporting Table 3: *Cl/(Mo+Ir) atomic ratio determined from XPS analysis. Values calculated for the most active materials in each series is reported in bold character.*

calcination temperature (°C)	Cl/(Mo+Ir)			
	10% Mo	30% Mo	50% Mo	pure Ir
400	0.32	0.56	0.17	0.38
450	0.14	0.17	0.12	0.21
500	0.05	0.03	0.04	0.20
550	0.03	0.02	0.06	0.13
600	0.03	0.02	0.04	0.15
800	0.05	0.02	0.03	

Temperature-resolved Fourier transform (FT) of the Ir L3-edge EXAFS spectra for pure Ir-sample and for the sample prepared with 30% and 50% Mo upon calcination are shown in the figure thereafter.

Supplementary Figure 11: (a) Temperature-resolved Fourier transform of the Ir L3-edge EXAFS spectra for pure Ir-sample upon calcination. Fourier transform of the Ir L3-edge EXAFS spectra for the sample prepared with (b) 30% Mo and (c) 50% and calcined at various temperatures.

Between 400 and 450 °C, the Ir-Cl peak at $R' = 1.99 \text{ \AA}$ (uncorrected for phase shift) disappears concomitantly with the appearance of the characteristic Ir-O peak at $R' = 1.65 \text{ \AA}$. From EXAFS fitting, at 400 °C, a number of Cl neighbor is higher than the oxygen neighbor (ca. 3.7 vs 2.1). At 450 °C, the trend inverses, and the number of O neighbor is predominant. It then reaches a constant value up to 800 °C. Whatever the Ir/Mo ratio, chloride content varies similarly. No distances characteristic of Ir-Cl bond is observed on EXAFS spectra above 450 °C. From XPS measurements, we also observe that Cl content strongly decreases from 400 °C to 450 °C (Table S3), and then reaches a constant value up to 800 °C of ca. 0.05 for the ratio Cl/(Ir+Mo). **The main conclusion is that highly active materials are obtained around the crystallization temperature with and without Cl.** The most active materials prepared without Mo and with 10% of Mo contain a rather high amount of residual chlorine. Inversely, the material prepared with 30% and 50% of Mo are converted into corresponding oxides (without Cl) at 500 °C, while most active material are observed at 500 °C and 550 °C for these two series, respectively. Therefore, the presence of chlorinated species cannot be accounted for explaining the higher/lower activity of the catalysts.

The second point regards lower oxidation state.

Here again, as shown in Figure 3a thereafter, the transition between Ir(III) and Ir(IV) occurs around 450 °C as determined by XANES and XPS. The materials prepared with 50% Mo and calcined at 500 °C present high oxidation number (+IV) but it is amorphous.

Part of Figure 3: (a) graph summarizing the main oxidation state observed from XPS spectra and XANES region spectra analysis for the different Mo content (theoretical %) as a function of calcination temperature.

In all the series, the most active materials are obtained around the crystallization temperature. The trend in electrochemical activity of IrMo mixed oxides is similar to pure iridium series, but is shifted towards higher temperature as the Mo content increases. Our hypothesis is that the reasons why we are observing these trends are similar for mixed oxides and pure iridium oxide. To support that we investigated separately each parameter that might play a role in the electrochemical activity. We are now clarifying this discussion in the paper by subdividing it into separate sections.

Actions taken:

New supporting Table: quantification of chlorine by fitting EXAFS spectra – Supporting Table 2.

New supporting Table: quantification of chlorine by XPS – Supporting Table 3.

New section: Influence of Mo and Cl

4. If you want to compare the performance of an electrochemically made amorphous oxide, activation process (0 ~ 1.5 VRHE, 500 mV s⁻², 100 cycles) was suggested. After activation process, amorphous catalyst surface would convert to electrochemically oxidized amorphous IrOx. However, crystalline IrO₂ almost maintains their state. The OER performance comparison of synthesized catalyst before and after activation process can offer information for debate of amorphous IrOx and crystalline IrO₂.

Reply: We thank the reviewer for the remark. Indeed electrochemical activation can be a useful method to distinguish amorphous from crystalline surfaces. In our case, we have decided to use X-ray based methods (XRD, XAS) that give direct information about the crystalline/amorphous character of our materials. XRD provides information on the crystallinity, the crystal structure and lattice parameters. In addition, considering the morphology of our catalysts (small nanoparticles in thin pore's walls), EXAFS describes carefully the local order around iridium and discriminate amorphous from crystalline materials.

Reviewer #3:

The revised version of the manuscript is improved. The authors have fixed a number of small errors and even included an in situ XAS measurement set. However, the data still does not support the conclusions.

The authors continue to claim "an optimal degree of ordering" can explain the activity of X-ray amorphous IrOx and rule out the accepted picture of iridium oxidation state as a factor. The data used for this claim remains inconclusive for a variety of reasons:

Reply: Thank you for your comments. Indeed, there are several parameters that could affect the activity of the

materials as listed by the reviewer thereafter. We are answering thereafter point by point to the parameters underlined by the reviewer; we have modified the structure of the paper to better clarify the discussion and adapted the conclusions accordingly and we believe that the paper has strongly been improved.

The samples show varying Cl content, a poison.

Reply: Thank you for the remark, this is an important point. There are several parameters that could affect the electrochemical activity and as state by the reviewer, the presence of chlorine could be one. However, we provide now in depth analysis that show that the presence of chlorinated species cannot be accounted for explaining the higher/lower activity of the catalysts.

To clarify this point, measured chlorine content by XPS for all the samples are reported, and, in addition we have now fitted EXAFS spectra of the samples prepared without Mo and with 30% and 50% Mo to determine the relative evolution of Ir-Cl and Ir-O with calcination. The average number of Cl and O in the coordination sphere of Ir from EXAFS spectra and the determined chlorine content as a function of the calcination temperature from XPS experiments are reported in Table S2 and Table S3, respectively.

Supporting Table 2a: Summary of the EXAFS fit Parameters for the pure IrO₂ calcined at different temperatures and recorded at the calcination temperature (except the 800 °C sample which was also recorded at RT.)^a

Sample	Shell	N	R / Å	$\sigma^2 / \text{Å}^2$	$\Delta E_0 / \text{eV}$	R _f (%)	χ^2_{red}
0% 400 °C	Ir-O	1.10 0.38	2.00 0.05	0.006 (fixed)	7.98 1.54	0.77	767.95
	Ir-Cl	4.44 0.33	2.35 0.01	0.005 (fixed)			
0% 450 °C	Ir-O	2.52 0.18	1.97 0.01	0.006 (fixed)	7.52 0.80	0.41	387.99
	Ir-Cl	3.08 0.16	2.34 0.01	0.006 (fixed)			
0% 500 °C	Ir-O	5.74 0.44	1.98 0.01	0.006 0.001	10.30 0.89	0.75	363.68
0% 550 °C	Ir-O	5.84 0.45	1.98 0.01	0.004 0.001	10.85 0.94	1.20	571.49
0% 650 °C	Ir-O	5.79 0.80	1.98 0.01	0.004 0.001	10.86 1.61	3.87	947.25
0% 800 °C	Ir-O	6 (fixed)	1.97 0.01	0.005 0.001	10.26 0.82	0.67	137.54
0% 800 °C (at RT) ^b	Ir-O	6 (fixed)	1.98 0.01 (DRX 2.00)	0.002 0.001	10.87 0.75	1.52	4442.11
	Ir-Ir ₁	2 (fixed)	3.14 0.01 (DRX 3.16)	0.002 0.001			
	Ir-O ₂	4 (fixed)	3.56 0.03 (DRX 3.41)	0.004 0.004			
	Ir-Ir ₂	8 (fixed)	3.54 0.01 (DRX 3.56)	0.001 0.001			

^a N, coordination numbers, r, distances between Ir and neighboring atoms, σ , Debye Waller factors, ΔE_0 , scaling energy parameter and R_f goodness of fit. Bold numbers correspond to fixed values and uncertainties are given in italic. Those parameters were obtained by least square fitting over a 1.0-3.0 Å range of the Fourier transforms of the experimental EXAFS spectra ($k_{\text{min}} = 2.8 \text{ Å}^{-1}$ and $k_{\text{max}} = 10.6 \text{ Å}^{-1}$, Hanning window with $dk = 1$). The reliability of the fit was assessed by the minimization of the reduced χ^2_{red} and metrics defined by the IXS standards and criteria committee (http://ixs.iit.edu/subcommittee_reports/sc/err-rep.pdf).

^b The fit of this sample was used to determine the value of the amplitude reduction factor $S_0^2 = 0.80$. The mean distances determined from the crystallographic structure for IrO₂ are reported for comparison purpose. Those parameters were obtained by least square fitting of over 1.1-4.0 Å of the Fourier transforms of the experimental EXAFS spectra ($k_{\text{min}} = 3.0 \text{ Å}^{-1}$ and $k_{\text{max}} = 14.0 \text{ Å}^{-1}$, Hanning window with $dk = 1$).

Supporting Table 2b: Summary of the EXAFS fit Parameters for the 30% Mo series calcined at different temperatures and recorded at RT.^a

Sample	Shell	N	R / Å	$\sigma^2 / \text{Å}^2$	$\Delta E_0 / \text{eV}$	R _f (%)	χ^2_{red}
30% 400 °C	Ir-O	2.11 0.53	2.02 0.02	0.003 0.002	10.13 0.83	1.76	1129.68
	Ir-Cl	3.80 0.49	2.35 0.01	0.004 0.001			
30% 450 °C	Ir-O	2.85 0.86	2.01 0.02	0.002 0.002	10.27 1.82	12.34	1179.96
	Ir-Cl	2.05 1.02	2.35 0.02	0.004 0.004			
30% 500 °C	Ir-O	5.56 0.37	1.99 0.01	0.002 0.001	10.44 0.87	4.21	1127.15
30% 550 °C	Ir-O	5.79 0.29	1.98 0.01	0.003 0.001	10.26 0.67	1.85	894.59
30% 600 °C	Ir-O	5.98 0.30	1.98 0.01	0.003 0.001	10.03 0.65	1.60	2079.70
30% 800 °C	Ir-O	5.89 0.29	1.98 0.01	0.002 0.001	10.64 0.64	1.54	2232.25

^a N, coordination numbers, r, distances between Ir and neighboring atoms, σ , Debye Waller factors, ΔE_0 , scaling energy parameter and R_f goodness of fit. Bold numbers correspond to fixed values and uncertainties are given in italic. Those parameters were obtained by least square fitting over a 1.0-3.0 Å range of the Fourier transforms of the experimental EXAFS spectra ($k_{\text{min}} = 2.8 \text{ Å}^{-1}$ and $k_{\text{max}} = 10.6 \text{ Å}^{-1}$, Hanning window with $dk = 1$).

Supporting Table 2c: Summary of the EXAFS fit Parameters for the 50% Mo series calcined at different temperatures and recorded at RT.^a

Sample	Shell	N	R / Å	$\sigma^2 / \text{Å}^2$	$\Delta E_0 / \text{eV}$	R _f (%)	χ^2_{red}
50% 400 °C	Ir-O	2.18 0.25	2.01 0.01	0.003 0.001	10.48 0.41	2.73	88.88
	Ir-Cl	3.73 0.21	2.35 0.01	0.004 0.001			
50% 450 °C	Ir-O	3.59 0.25	2.00 0.01	0.005 0.001	10.47 0.35	2.69	159.42
	Ir-Cl	2.66 0.18	2.35 0.01	0.003 0.001			
50% 500 °C (Oxygens only)	Ir-O	5.76 0.24	1.99 0.01	0.003 0.001	10.87 0.53	1.35	851.11
50% 500°C	Ir-O	5.60 0.32	1.99 0.01	0.003 0.001	10.79 0.59	1.12	993.43
	Ir-Cl	0.15 0.21	2.34 0.05	0.001 0.001			
50% 550 °C	Ir-O	6.03 0.28	1.98 0.01	0.003 0.001	10.53 0.60	1.35	1453.51
50% 600 °C	Ir-O	5.76 0.31	1.99 0.01	0.002 0.001	10.94 0.70	1.52	1027.26
50% 800 °C	Ir-O	5.90 0.27	1.98 0.01	0.002 0.001	10.74 0.60	1.19	281.97

^a N, coordination numbers, r, distances between Ir and neighboring atoms, σ , Debye Waller factors, ΔE_0 , scaling energy parameter and R_f goodness of fit. Bold numbers correspond to fixed values and uncertainties are given in italic. Those parameters were obtained by least square fitting over a 1.0-3.0 Å range of the Fourier transforms of the experimental EXAFS spectra ($k_{\text{min}} = 2.8 \text{ Å}^{-1}$ and $k_{\text{max}} = 10.6 \text{ Å}^{-1}$, Hanning window with $dk = 1$).

Supporting Table 3: Cl/(Mo+Ir) atomic ratio determined from XPS analysis. Values calculated for the most active materials in each series is reported in bold character.

calcination temperature (°C)	Cl/(Mo+Ir)			
	10% Mo	30% Mo	50% Mo	pure Ir
400	0.32	0.56	0.17	0.38
450	0.14	0.17	0.12	0.21
500	0.05	0.03	0.04	0.20
550	0.03	0.02	0.06	0.13
600	0.03	0.02	0.04	0.15

Temperature-resolved Fourier transform (FT) of the Ir L3-edge EXAFS spectra for pure Ir-sample and for the sample prepared with 30% and 50% Mo upon calcination are shown in the figure thereafter.

Supplementary Figure 11: (a) Temperature-resolved Fourier transform of the Ir L3-edge EXAFS spectra for pure Ir-sample upon calcination. Fourier transform of the Ir L3-edge EXAFS spectra for the sample prepared with (b) 30% Mo and (c) 50% and calcined at various temperatures.

Between 400 and 450 °C, the Ir-Cl peak at $R' = 1.99 \text{ \AA}$ (uncorrected for phase shift) disappears concomitantly with the appearance of the characteristic Ir-O peak at $R' = 1.65 \text{ \AA}$. From EXAFS fitting, at 400 °C, a number of Cl neighbor is higher than the oxygen neighbor (ca. 3.7 vs 2.1). At 450 °C, the trend inverses, and the number of O neighbor is predominant. It then reaches a constant value up to 800 °C. Whatever the Ir/Mo ratio, chloride content varies similarly. No distances characteristic of Ir-Cl bond is observed on EXAFS spectra above 450 °C. From XPS measurements, we also observe that Cl content strongly decreases from 400 °C to 450 °C (Table S3), and then reaches a constant value up to 800 °C of ca. 0.05 for the ratio Cl/(Ir+Mo).

The main conclusion is that **highly active materials are obtained always around the crystallization temperature with and without Cl**. The most active materials prepared without Mo and with 10% of Mo contain a rather high amount of residual chlorine. Inversely, the material prepared with 30% and 50% of Mo are converted into corresponding oxides (without Cl) at 500 °C, while most active material are observed at 500 °C and 550 °C for these two series, respectively. Therefore, the presence of chlorinated species cannot be accounted for

explaining the higher/lower activity of the catalysts. Another example, for instance, in the 50% Mo, samples calcined at 500, 550 and 600 present similar Mo and Cl contents, but the most active is the one calcined at 550 °C.

Actions taken:

New supporting Table: quantification of chlorine by fitting EXAFS spectra – Supporting Table 2.

New supporting Table: quantification of chlorine by XPS – Supporting Table 3.

New section: Influence of Mo and Cl

It is not clear how the active surface area is changing between samples.

Reply: Indeed, this is a very important parameter to be discussed. As mentioned in the previous version of the answer to reviewers, measuring the electrochemical active surface area of Ir-based catalyst is a strong unresolved challenge. Iridium oxide electrochemical surface area cannot be probed by usual CO stripping experiments for instance.

To investigate the surface area evolution we have performed physisorption measurements for all the samples. The value of surface areas and crystallites size (determined from XRD) are reported in supporting Figure S15 and supporting Table 4, respectively. In each series, we observe that the surface area increases up to the crystallization temperature. Indeed the beginning of the crystallization corresponds to formation of small nanoparticles characterized by a high surface area. Increasing the calcination temperature leads to crystal growth that is usually accompanied by a decrease of surface area. This decrease is clearly observed for Ir and IrMo 10% characterized by a larger crystal sizes. However, the surface area of IrMo materials 30% and 50% does not vary and stabilize at ca. $70 \text{ m}^2 \text{ g}^{-1}$; this is due to the formation of mesopores within the walls which counterbalances the decrease in surface area that should be observed due to particle growth as shown in the BET isotherms in Figure S15. While the increase of the activity between amorphous and crystalline materials could be attributed to an increase in surface area, the decrease in activity observed between very small crystallites and larger ones cannot be attributed only to the variation in surface area. This difficulty in correlating surface area and activity is not completely unexpected. Indeed, it has been shown in the literature that atoms participating in the reaction are not only the one on the surface. Bulk atoms can participate but the depth at which atoms are involved depend on the crystalline state of the material. One recent study shows that for rutile IrO_2 , atoms down to 2.5 nm are participating, while deeper atoms can participate in amorphous materials (10.1021/acs.jpcclett.0c01258). According to these studies, in our samples, the crystallite sizes are so small that all atoms could be involved in the reaction making any correlation between BET and electroactivity measurement unreliable. According to this, this aspect now discussed in the paper.

Action taken:

New section: Influence of surface area and crystallinity.

New supporting figure: supporting Figure 15 on surface area measurements.

New supporting Table: supporting Table 4 for crystallite size.

Mo oxidation state changes and correlates with activity.

Reply: Indeed, as stated in the paper and supported by the XPS spectra, Mo oxidation changes in a certain range of temperature, however Mo oxidation state does not correlate with activity.

As mentioned in the paper, for all the samples studied, Mo(VI) is observed for materials calcined at 400 °C. For all the samples, whatever the Mo content, Mo(VI) is reduced into Mo(V) between 400 and 500 °C. At 500 °C, 550 °C and 600 °C, whatever the Mo content, we only detect Mo(V). For all the samples, at 800 °C, very a small contribution of Mo(IV) is observed along with Mo(V) (Figure S8). From the experimental results, we can say that the most active materials prepared with 10% contains a mixture of Mo(VI) and Mo(V) species, while the most active materials prepared with 30% and 50% Mo contains only Mo(V) species.

Another example, it that the electroactivity of $\text{Ir}_{0.5}\text{Mo}_{0.5}\text{O}_x$ calcined at 500 °C is lower than that of the same material calcined at 550 °C, while they present similar Mo oxidation state.

From these analyses, we do not observe a correlation between the Mo oxidation state and the electrochemical activity.

It is not clear what the ratio of Mo/Ir on the surface is or if it is changing.

Reply: Thank you, this aspect need to be clarified. In order to estimate the Mo/Ir content at the surface we have probed the Ir and Mo content by XPS that is a surface sensitive technique. It is important to precise that our samples are typically composed nanoparticles with sub-10 nm size that is similar to the penetration depth of the technique. Consequently, these absolute values must be considered with care. Looking at the results in the table hereafter we can see that the amount of Mo% of each series is generally in agreement with the theoretical composition and, most importantly, doesn't vary significantly with the calcination temperature. This observation supports the results obtained by XRD and EDX that excluded a temperature-induced phase separation / segregation of Ir on the surface (except for the sample indicated with a *, as already discussed in the article).

Mo/(Mo+Ir) atomic percentage determined from XPS measurements. Most active materials in each series is reported in bold character.

Theoretical Mo content	10%	30%	50%
Calcination temperature	% Mo XPS	% Mo XPS	% Mo XPS
400	12	46	52
450	13	34	62
500	16	34	48
550	15	35	49
600	12	35	50
800	17	31	38*

*Mo content decreases because for the 50% Mo sample, at 800 °C the Mo content decreases down to 38% to form $\text{Ir}_{0.6}\text{Mo}_{0.4}\text{O}_x$ compound as determined from XRD.

Actions taken:

We integrated these values in supporting Table 1.

Mo introduces variations in the lattice parameters.

Reply: We agree, the reviewer is right, Mo induces variation in lattices parameters and this could have an influence on the activity. In the previous version of the paper, it was mentioned that Mo induces variation in lattice parameter: "Compared to pure IrO_2 , the position of the (110) peak shifts towards low 2θ values as the Mo content increases in the IrMo-mixed oxide while the (011) peak shifts towards higher 2θ . This is indicative of an increase in the a unit-cell parameter and a concomitant decrease in the c parameter as illustrated in Figure S10."

But now, we are providing an exhaustive view of a and c parameters, we have now calculated these parameters for each samples from XRD diffractograms for crystalline materials using MAUD software; values are reported in the figure thereafter and are now included as a new supporting Figure (S14). From this graph, we can observe that both a and c parameters weakly vary with temperature in each series. The introduction of Mo has a weak influence on the a parameter (except for the material prepared with 50% Mo and calcined at 550 °C, however, uncertainties on this particular value can be due to the small particle size and large XRD peak). The c parameter decreases of ca. 0.8 Å between pure IrO_2 and the sample prepared with 50% Mo. When the sample containing 50% Mo is calcined at 800°C, we find a and c parameters equal to that of the sample prepared with 30%, which confirms the XRF results indicating the decrease of Mo content from 50% down to 30% upon calcination between 600 and 800 °C.

Supplementary Figure 14: Evolution of a and c parameters as a function of the calcination temperature.

Considering these data, for each series, the a and c parameters does not vary significantly with the calcination temperature while the activity is strongly varying. Sun et al. (DOI: 10.1039/c7cc09580e) have reported that the lattice strain decreases the c/a ratio. If we consider the c/a as a representative value of the lattice strain, in our sample the c/a ratio remains constant for pure IrO_2 , and the sample prepared with 10% of Mo from 450 °C to 800 °C. It stays constant between 500 and 600 °C for 30% Mo, and is slightly higher at 800 °C. The c/a ratio increases with temperature for the sample prepared with 50% Mo, however, strong uncertainties due to very small particle sizes as well as the decrease of Mo content at 800 °C render measurement on this series less reliable. **Overall, in each series, the lattice strain does not seem to vary strongly with temperature while activity is changing.**

Action taken:

We had a new supporting Figure: Variation of a and c parameters with temperature – supporting Figure 14. We also discuss this trend on the manuscript in the section "Influence of surface area and crystallinity".

It is not clear if different facets are exposed upon alloying/annealing.

Reply: This aspect needs to be clarified. Most of the materials reported in this work are amorphous or contains ultrasmall crystallites (size < 4nm). TEM images in the new Figure 1 and new Figure 4 show that the particles are mostly round-shape and as a consequence do not present preferentially exposed facets. Large particles are more faceted (especially in materials without or with a low % of Mo) but crystals are randomly oriented in the whole volume of the catalysts as shown in the TEM micrographs of IrO_2 shown hereafter. For these reasons, we can exclude that the some facets are exposed preferentially.

However, our initial description of the material was maybe not clear enough and we have remedied to this by adding a more detailed description of the material (see question below).

Actions taken:

New figure: description of the morphology of the materials – Figure 1.

New section: detailed description of the material morphology including discussing this crystal shape and facets.

All of these could be influencing activity. It has been established that, for instance, surface termination/average surface oxidation state influences activity, see ACS Catal. 2020, 10, 806–817 and J. Phys. Chem. Lett. 2014, 5, 1636–1641. And while such variations are quantifiable, it is not clear what "an optimal degree of ordering" is. The authors should quantify ordering and show the variations expected to influence activity do not, e.g. area, faceting, oxidation state, strain.

Reply: The reviewer is right, the term "optimal degree of ordering" is unclear, and we have removed it from the text. Instead, we believe that particles size is a parameter that can be clearly quantified.

As mentioned in the previous answers indeed the structure is crucial however some other parameters needs to be considered. We have discussed, the possible effect of Ir oxidation state, surface area and facets in the previous answers and two dedicated sections were added in the revised version of the manuscript.

As suggested by the reviewer, activity can be correlated with surface distortion/termination as for the two papers mentioned in reviewer comment. The reviewer also mentioned strain. A possible effect of distortion or strain is not in contradiction with our conclusions however the quantification of these parameters is challenging. Indeed, these previous studies were performed on model materials prepared as thin films by physical deposition methods (*Epitaxially grown oriented (100) iridium oxide films of 45 ± 3 nm for ACS Catal. 2020, 10, 806–817; 25 nm thick films obtained by PLD for ref. J. Phys. Chem. Lett. 2014, 5, 1636–1641*). While characterization of controlled thin films can give valuable information, this quantification can't be done reliably on powder materials such as our materials.

Instead, as mentioned previously, Sun et al DOI: 10.1039/c7cc09580 used crystal parameters (c/a ratio) to quantify the change in lattice strain of IrO_x by a simple equation. As shown in suppl. Figure S14 the the a and c parameters does not vary significantly with the calcination temperature while the activity is strongly varying.

Actions taken:

This aspect is discussed now in the section " influence of surface area and crystallinity"; we added an additional reference on lattice strain DOI: 10.1039/c7cc09580

The authors still use principally ex situ characterization techniques, and there is value in correlating ex situ results with activity. But the authors present no evidence any such correlation is anything but accidental.

The *in situ* XAS, while potentially useful, is not analyzed in depth and was only performed for one sample. As the authors note, this technique is bulk sensitive and, for the large particles used in this work, will show only minor changes due to the expected differences in surface oxidation during operation.

Reply: This comment is very similar to reviewer 2 and we realized that we failed in well describing our material. As mentioned earlier, we do believe that there was a misunderstanding of our material structure since it was not properly described. The presented XAS data were actually performed on a series of materials, *i.e.* 3 different materials, especially designed for maximizing the number of Ir atoms involved in the OER as described thereafter. Therefore, we observe a significant shift in the peak-top position since we observe an increase of 1.1 eV for the sample calcined at 450 °C and of 0.5 eV for samples calcined at 500 °C and 550 °C. These values are comparable with those reported by Strasser and co-workers in ref. 10.1038/s41929-018-0153-y.

We have now added a whole paragraph at the beginning of the paper and a new figure to fully describe the materials and explain why *in situ* and *ex situ* XAS measurements are relevant.

The new Figure is shown thereafter.

Figure 1: (a) Schematic illustration of the synthesis process for the formation of porous $\text{Ir}_x\text{Mo}_{(1-x)}\text{O}_y$ catalysts represented in the TEM micrograph (right side). SEM micrographs sequence of IrO_2 , $\text{Ir}_{0.9}\text{Mo}_{0.1}\text{O}_2$, $\text{Ir}_{0.7}\text{Mo}_{0.3}\text{O}_2$ and $\text{Ir}_{0.5}\text{Mo}_{0.5}\text{O}_2$ porous catalysts obtained at 500 °C. (c) Illustration of the morphology of a typical nanocrystalline catalyst. (d) and (f) HR-TEM micrographs of a single pore and of nanocrystalline wall respectively. (f) STEM image and corresponding EDX analysis of a catalysts containing 30% of Mo and calcined at 550 °C: (h) Mo L edge (green), (i) Ir M edge (red) and (g) overlapping of the two elements.

The morphology of a typical mixed-oxide catalyst is thus illustrated in Figure 1(c): the microspheres are highly porous (typically 75% vol.) composed of an interconnected macroporosity with thin walls that can be amorphous or composed by isotropic nanocrystals depending on the calcination temperature.

Each macropores has a size of ca. 250 nm. The **walls of the porous microsphere are very thin**, typically below 15 nm, as determined by HR-TEM. The pore's **walls are composed only of a few layers of crystalline nanoparticles** in the case of crystalline materials. In most cases, the particles are very small. The crystallite size were displayed in the main paper (currently Figure 4a), but for the sake of clarity, we are also reporting numerical values in Table S4.

Supporting Table 4: Crystallite size determined by Rietveld refinement of the corresponding X-ray diffractograms as a function of calcination temperature. Most active materials in each series is reported in bold character. *The crystallite size determined for pure IrO_x materials is an average between size calculated from a and from c parameters. (see ref. 10.1002/aenm.201802136 for details).

Calcination temperature (°C)	Crystallite size (nm)			
	10% Mo	30% Mo	50% Mo	0% Mo
400	n.a.	n.a.	n.a.	n.a.
450	1.5	n.a.	n.a.	8.6*
500	6	2.5	n.a.	10*
550	7	4	2	12.5*
600	7.5	4.5	2.5	15.8*
800	11	7	5	27.7*

Therefore, even if the point raised by the reviewer about the accuracy of XAS in detecting the oxidation state in relation to the crystalline properties is very pertinent, the concept of bulk techniques becomes less relevant in the case of our materials. Indeed, the material was designed with this morphology because it is ideally suited for fundamental studies also by bulk sensitive techniques. The high accessibility of the porosity and the thin walls lead to a maximization of the Ir-based material involved in the catalytic reactions. This allow probing both surface and core Ir-based materials by XAS during operando experiments.

This can be confirmed experimentally by XAS. To clarify that, we have modified Figure 3: we have merged the discussion about *ex situ* and *in situ* XAS measurements, we have included additional SEM micrograph of the samples, and we have re-written the paragraph on iridium oxidation state. Part of this new Figure is presented thereafter.

series 30% Mo

Part of Figure 3: (b-d) SEM images of the series prepared with 30% Mo and calcined at 450 °C, 500 °C and 550 °C and (e-f) corresponding operando characterization at the Ir L3-edge: (e) XANES spectra recorded at 0.2 V vs. NHE, 1.7 V vs. NHE and back to 0.2 V vs. NHE and the corresponding peak-top values; (f) Superposition of the EXAFS spectra recorded for the most electrochemically active materials of the 30% Mo series (calcined at 500 °C) at 0.2 V vs. NHE (black diamonds), 1.7 V vs. NHE (pink circles) and back to 0.2 V vs. NHE (blue triangles).

The new SEM micrographs of the three samples displays that the materials have similar morphology, but while the sample calcined at 450 °C is amorphous, the samples calcined at 500 °C and 550 °C exhibit crystalline pore's walls. From these SEM micrographs, we believe we better highlight the peculiar porous morphology with thin pore's wall.

This allows for probing the iridium atoms located at the outer surface and in the inner porosity by *in situ* XAS experiments.

XANES spectra of the materials recorded at various potential and corresponding peak top values are plotted on Figure 3e. The initial XANES peak top positions measured in the electrochemical cell at 0.2 V vs. NHE confirmed *ex situ* measurements: the oxidation state is lower (corresponding to Ir(III)+Ir(IV)) for materials calcined at 450 °C and increases to Ir(IV) for materials calcined at 500 °C and 550 °C. At 1.7 V vs. NHE, under OER conditions, as expected, the oxidation state of all the materials increases. More importantly, whatever the initial oxidation state, all of them tends to have a similar oxidation state during OER (> Ir(IV)). This higher oxidation state (formally >+4) observed at high potential is in agreement with results from the literature where it has been shown that the Ir oxidation state can increase at high potential (10.1021/acs.jpcclett.8b00810, 0.1038/s41929-018-0153-y, 10.1039/C0CP01472A). Since the XAS

experiment probes all the Ir atoms of the material, this significant increase in XANES peak top position (similar to a previous study *Nat. Cat.* **1**, 841-851, 2018) is an additional evidence that the reaction is not occurring only at to outer surface of the micrometric spheres but also it also inside the microspheres. This would not be the case if the core iridium materials were not accessible as in the case of not porous materials.

Indeed, in light of these results, we can confirm that whatever the initial oxidation state of our catalysts, during OER the Ir oxidation state increases toward similar values. This implies that the differences in electrochemical activity measured for the 30% Mo series (calcined at 450 °C, 500 °C and 550 °C) do not depend on the oxidation state during OER. In addition, from EXAFS spectra (Figure 3f) the reversibility of the process indicates that there is no major structural or chemical changes after OER. The experiments performed both *ex situ* and *in situ*, on three different samples, indicate that the electroactivity trend shown in cannot be correlated with initial iridium oxidation state, nor the one measured *operando*.

Actions taken:

New Figure: Figure 1

Modification of Figure 3

New section: to better describe the formation/morphology of our materials

Modified section: possible influence of iridium oxidation state

New table: crystallite size, supporting Table 4

If the authors are able to show ordering mediates rate, an atomic scale mechanism would be helpful. That is, how does ordering change the rate?

Reply: As mentioned above, the term “optimal degree of ordering” has now been removed from the text. Instead, we believe that crystal size is a parameter that better enables the quantification of the local degree of order. Notably, we show that all the most active samples shows a similar characteristic EXAFS spectrum, that is a practical way to describe how an "optimal structural feature" should look like. In the discussion, we also pointed out that this characteristic EXAFS curve was observed in highly active materials prepared by other groups. That is to say that the objective of the paper is to provide experimental evidences allowing identifying the most pertinent parameters to design highly active electrocatalysts. We believe that investigating the atomic scale OER mechanism on these complex materials would be very valuable for the community but it would better suited for a dedicated study.

I also found the discussion of the material confusing. The authors claim to have Ir_{0.5}Mo_{0.5}O₂. From XPS Mo is Mo(VI) and Mo(V) and Ir is Ir(IV). This cannot be Ir_{0.5}Mo_{0.5}O₂. The authors should clarify this issue.

Reply: Thank you for this comment. Indeed, the exact stoichiometry of O to metal atoms can't be defined with certitude as in most iridium oxides materials reported in the literature. Consequently, we have therefore replaced all labels by Ir_xMo_(1-x)O_y.

REVIEWERS' COMMENTS

Reviewer #2 (Remarks to the Author):

The Authors well replied to all my remarks and provided good supported data. I think that this manuscript is acceptable for publication. However, my only concern is that readers confuse that the oxidation state of all iridium materials is not related to OER performance. In conclusion, authors well comments that "We therefore propose that the high electrochemical activity of iridium-based oxides calcined at low temperature is unlikely due to the iridium oxidation state, but rather strongly depends on the material structure." My only minor suggestion is that last sentence of the abstract should be changed to something like below.

"This study indicates that short-range ordering, corresponding to sub-2 nm crystal size for our samples, drives the activity independently of the initial oxidation state and composition of the calcined iridium oxides."

Reviewer #3 (Remarks to the Author):

The authors have done a nice job of addressing my concerns and now present compelling evidence of the importance of short range order on OER activity of Ir based oxides independent of initial oxidation state. I expect the methods and findings will spur new avenues of investigation in OER on Ir and other materials.

A minor point. The schematics in Figure1a and 1c could be improved a bit. In the version I have there are large white boxes cutting through the Ir and Mo spheres in 1a and white boxes around the arrows in 1c.

REVIEWERS' COMMENTS

Reviewer #2 (Remarks to the Author):

The Authors well replied to all my remarks and provided good supported data. I think that this manuscript is acceptable for publication. However, my only concern is that readers confuse that the oxidation state of all iridium materials is not related to OER performance. In conclusion, authors well comments that "We therefore propose that the high electrochemical activity of iridium-based oxides calcined at low temperature is unlikely due to the iridium oxidation state, but rather strongly depends on the material structure." My only minor suggestion is that last sentence of the abstract should be changed to something like below.

"This study indicates that short-range ordering, corresponding to sub-2 nm crystal size for our samples, drives the activity independently of the initial oxidation state and composition of the calcined iridium oxides."

Answer to reviewer #2:

We would like to thank the reviewer for her/his comments. We have modified the last sentence of the manuscript as suggested by the reviewer.

Reviewer #3 (Remarks to the Author):

The authors have done a nice job of addressing my concerns and now present compelling evidence of the importance of short range order on OER activity of Ir based oxides independent of initial oxidation state. I expect the methods and findings will spur new avenues of investigation in OER on Ir and other materials.

A minor point. The schematics in Figure1a and 1c could be improved a bit. In the version I have there are large white boxes cutting through the Ir and Mo spheres in 1a and white boxes around the arrows in 1c.

Answer to reviewer #3:

We would like to thank the reviewer for her/his comments. We have checked the images and believe that it was a pdf processing error. It will be corrected in the new version.